# `DecompSR`: A Dataset for Decomposed Analyses of Compositional Multihop Spatial Reasoning

**Lachlan McPheat**      *lmcpheat@turing.ac.uk*
**Navdeep Kaur**      *nkaur@turing.ac.uk*
*The Alan Turing Institute*

**Robert E. Blackwell**      *r.e.blackwell@leeds.ac.uk*
*The University of Leeds*

**Alessandra Russo**      *arusso@turing.ac.uk*
*The Alan Turing Institute*
*Imperial College London*

**Anthony G. Cohn**      *a.g.cohn@leeds.ac.uk*
*The Alan Turing Institute*
*The University of Leeds*

**Pranava Madhyastha**      *pmadhyastha@turing.ac.uk*
*The Alan Turing Institute*
*City St George's University of London*

**Reviewed on OpenReview:** *https://openreview.net/forum?id=P81p2nTuvA*

## Abstract

We introduce `DecompSR`, decomposed spatial reasoning, a large benchmark dataset (over 5m datapoints) and generation framework designed to analyse compositional spatial reasoning ability. The generation of `DecompSR` allows users to independently vary several aspects of compositionality, namely: productivity (reasoning depth), substitutivity (entity and linguistic variability), overgeneralisation (input order, distractors) and systematicity (novel linguistic elements). `DecompSR` has been built procedurally in a manner which makes it is correct by construction, which is independently verified using a symbolic solver to guarantee the correctness of the dataset. `DecompSR` is comprehensively benchmarked across a host of Large Language Models (LLMs) where we show that LLMs struggle with productive and systematic generalisation in spatial reasoning tasks whereas they are more robust to linguistic variation. `DecompSR` provides a provably correct and rigorous benchmarking dataset with a novel ability to independently vary the degrees of several key aspects of compositionality, allowing for robust and fine-grained probing of the compositional reasoning abilities of LLMs.

## 1 Introduction

Large Language Models (LLMs) are increasingly tasked with complex problem-solving, the capacity for systematic reasoning, i.e., the ability to productively apply learned rules and components to novel combinations and situations, has become an important concern, distinguishing genuine inferential capability from surface level pattern matching (Fodor & Pylyshyn, 1988; Bahdanau et al., 2019). Many contemporary applications, especially in scientific and logical domains, demand robust, generalisable inference (Bubeck et al., 2023; Trinh et al., 2024). However, despite emergent proficiency on diverse tasks (Huang et al., 2023), the true depth of LLM systematicity, especially their ability to generalise compositionally, remains a critical concern

and a central challenge for the machine learning community (Lake & Baroni, 2018; Srivastava et al., 2023). While LLMs demonstrate emergent proficiency across diverse tasks (Huang et al., 2023), their capacity for consistent systematic generalisation, particularly in the face of compositional novelty, remains an active area of investigation and concern (Dziri et al., 2023; Peng et al., 2024; Thomm et al., 2024; Fodor et al., 2025).

Current dominant evaluation paradigms focus on final-answer correctness, often obscuring underlying deficits in systematicity, and making it difficult to discern if models are truly competent. This narrow focus inadvertently leads to rewarding 'shortcut learning' effects where models exploit first-order statistical cues within benchmarks and the vast training resources, rather than engaging in the systematic composition of information (McCoy et al., 2023). Such evaluations rarely probe for productivity (generalising rules to more complex instances) or invariance to superficial changes, leading to models that are proficient on familiar data but brittle when facing novel problems demanding genuine rule-based generalisation (Liu et al., 2023; Xu et al., 2024). Further complicating assessment, directly scrutinising the reasoning process is also seemingly difficult. While 'chains-of-thought' (Wei et al., 2022; Prystawski et al., 2023; Guo et al., 2025) offer glimpses of reasoning processes, verifying the true systematicity of natural language justifications is resource-intensive and often inconclusive regarding principled generalisation (Nguyen et al., 2024; Lee & Hockenmaier, 2025; Marjanovic et al., 2025). At the same time, systems which allow for formal verification struggle with the richness of real-world inputs, and mechanistic interpretability (Elhage et al., 2021; Olsson et al., 2022; Nanda et al., 2023) is yet to reliably identify the specific neural circuits underpinning systematic generalisation, especially in complex domains like spatial reasoning where the systematic construction and manipulation of mental models are key but poorly measured for LLMs (Li et al., 2023; Cheng et al., 2024), as opposed to the rich literature on mental models in cognitive science (Craik, 1943; Johnson-Laird, 1983).

To address these critical gaps in evaluating compositional spatial reasoning, we introduce `DecompSR` (Decomposed Spatial Reasoning). We construct a generative framework to create task instances that systematically probes distinct facets of compositionality, as delineated by Hupkes et al. (2020). The core idea is that a system demonstrating consistent and correct performance across these systematically varied instances — designed to test *systematicity*, *productivity*, *substitutivity*, and *overgeneralisation* independently — is more likely to be employing robust, generalisable spatial reasoning processes rather than superficial heuristics. `DecompSR` achieves this by allowing precise control over parameters such as the number of reasoning steps (or hops – $k$), the linguistic expression of spatial relations, the nature of entities, and the presence of distracting information or structural disorder in the problem presentation. This controlled variation aims to neutralise cues that facilitate shortcut learning, forcing models to engage with the deeper compositional structure of spatial problems. Since LLMs see copious amounts and types of text including natural language spatial descriptions during pre-training, fine tuning and post-training, it is perhaps only logical to first focus on the predominant modality of LLM's training as a method to evaluate the compositional abilities of LLMs.

The core focus of `DecompSR` is to provide a reasoning task which stress-tests compositional abilities. It is desirable for an automated reasoning system to be robust in its reasoning even under the stress of long reasoning-chains, novel components, systematically altered entity names and so on. `DecompSR` provides a flexible and robust dataset to evaluate these abilities to extreme lengths. Our primary contributions are: a) a novel methodology and open-source framework for generating correct-by-construction natural language spatial reasoning tasks that enable decomposed evaluation of compositional abilities beyond mere accuracy; b) the release of `DecompSR`, a large-scale, customisable benchmark dataset (over 5 Million samples) for multi-hop compositional spatial reasoning. Our secondary contributions are: c) benchmarking of contemporary LLMs, revealing specific strengths and weaknesses in their systematic reasoning capabilities; and d) empirical findings demonstrating that while LLMs show some linguistic resilience, they largely struggle with systematic and productive generalisation in spatial tasks and exhibit overgeneralisation tendencies.

## 2 Background

### 2.1 On Reasoning Benchmarks

The improving capabilities of LLMs has driven significant progress in benchmarks that are devised to measure complex processes like reasoning. The field has seen the development of broad benchmark suites like ARC-

AGI (Chollet, 2019) and BIG-Bench (Srivastava et al., 2023), along with datasets focused on specific domains such as mathematical problem-solving (e.g., GSM8K (Cobbe et al., 2021), MATH (Hendrycks et al., 2021b)) or commonsense and logical inference (Talmor et al., 2019; Clark et al., 2020; Lin et al., 2025; Liu et al., 2025). Despite their utility, a prevailing limitation of many such benchmarks is their primary reliance on final-answer correctness. This focus can obscure whether a model has truly engaged in robust reasoning or has instead exploited superficial 'shortcut' strategies tied to patterns within the benchmark data (McCoy et al., 2023). Furthermore, concerns about potential data contamination - the inadvertent inclusion of test items in large-scale training sets-complicate the interpretation of reported performance (Sainz et al., 2023; Balloccu et al., 2024). In response, there is a discernible shift towards evaluation paradigms offering greater control. Procedurally generated (synthetic) datasets allow for precise manipulation of task features, facilitating more systematic investigation of specific reasoning abilities (Hendrycks et al., 2021a). Hybrid methods, which combine synthetic structures with natural language phrasing, aim to balance this control with linguistic information to mimic real structures, as seen in benchmarks like (Sinha et al., 2019; Shi et al., 2022; Mirzadeh et al., 2025). These approaches signify progress towards assessing the process of reasoning, not merely its outcome. Our work is based on the latter directions, and exploits synthetic generation where we are able to control several parameters for measuring systematicity.

## 2.2 Spatial Reasoning

The domain of spatial reasoning offers a particularly advantageous setting for investigating compositional generalisation. The inherent structure of spatial problems involving entities, their relations, and transformations, allows for the precise construction of test scenarios where the ground truth is unambiguous. Broadly speaking, spatial reasoning involves reasoning about spatial qualities such as directions, positions, connections and relations between objects. For a richer introduction to theories of spatial reasoning we recommend survey papers such as (Chen et al., 2015), covering topics like the Region Connection Calculus, various types of direction relations and calculi describing them, connections between absolute and relative directionality and more. Spatial information, and in particular qualitative spatial information is ubiquitous in text and being able to reason about it is essential if computers are to be able to process it effectively (Hayward & Tarr, 1995). For example, consider the Corpus of Lake District Writings (Rayson et al., 2017) a three-century long corpus of travel writings including works by Coleridge and Wordsworth which has rich spatial information in it. Equally, being able to reason about spatial information is key to understanding the everyday physical world, which humans (and robots) inhabit and perform actions located in time and space. Being able to communicate about spatial information underlies this, and indeed much of common sense, e.g. see Hayes' Naive Physics Manifesto (Hayes, 1985). Further there is research demonstrating how linguistic spatial relations influences learning (Carlson & Logan, 2005; Gilligan et al., 2017). There is also a growing interest in studying the internal representations of spatial information in LLMs as studied in (Wu & Deng, 2025).

Early benchmarks such as (Lake & Baroni, 2018) evaluating the ability of neural models to understand how to compose instructions in natural language and translate them into chains actions and (Ruis et al., 2020) grounding the previous dataset in a 2D grid, were pivotal in assessing how well models generalise from simple linguistic commands to new sequences of actions. These primarily probed systematicity: the capacity to correctly interpret and use familiar components (like primitive actions or object types) in previously unseen combinations. Subsequent work, including (Shi et al., 2022) and extensions such as (Li et al., 2024), shifted the focus towards multi-step relational inference from textual descriptions of paths or object arrangements. These benchmarks aimed to test *productivity*: the ability to apply learned inferential rules across instances of increasing length or complexity. This was often done through *k-hop* (sometimes called 'multi-hop') reasoning problems, where $k$ denotes the number of explicit relational statements, or 'hops' that must be chained together to deduce the relationship between two queried entities. However, the work of Shi et al. (2022) contains a structural error which makes $k$ an upper bound of the number of steps needed to solve a problem instance. We remedy this in `DecompSR`, ensuring $k$ is a crisp measure of the number of reasoning steps needed.

### 2.3 Compositionality

The limitations in assessing systematicity and productivity, as discussed previously, highlight the need to consider compositionality—the principle that complex meanings arise from constituent parts and their combination rules (Fodor & Pylyshyn, 1988; Partee, 2008), stemming from theories in linguistics (Frege et al., 1892) and logic (Boole, 1854). For AI systems like LLMs, true compositional understanding means flexibly combining learned knowledge for novel instances across diverse structural and semantic variations, going beyond basic systematicity (novel combinations of known parts) and productivity (scaling with complexity).

While LLMs have demonstrated strong performance on low-level natural language tasks, the limits of their compositional reasoning capabilities remains unclear. Neuro-symbolic approaches like (Olsson et al., 2022; Yang et al., 2023; Li et al., 2024) have shown promising steps to combine the strengths of LLMs with compositional reasoning of symbolic systems. These methods hinge on translating natural language into a structured formal language, which has further spurred the development of probabilistic logics (Yang et al., 2020; Manhaeve et al., 2021) which allow for a closer integration of neural and symbolic methods.

A seminal framework for a more fine-grained analysis of such compositional skills was provided by Hupkes et al. (2020) which argued that evaluating compositionality directly on natural language is challenging due to its inherent complexity and confounding factors. They introduced PCFG SET which is an artificial translation task using probabilistic context-free grammars, ensuring that compositionality was a salient and necessary to successfully translate the PCFG SET data. This allows for evaluation of different facets of compositionality, including systematicity, productivity, substitutivity, and overgeneralisation.

`DecompSR` draws inspiration from this decomposed evaluation strategy (hence the choice of name), but while PCFG SET offers a powerful tool for analysing models trained on its specific synthetic structures, its abstract *language* is perhaps not what LLMs usually encounter during their extensive pre-training. `DecompSR`, in contrast, is entirely grounded in natural language spatial reasoning, allowing us to probe how LLMs apply (or fail to apply) compositional skills in a familiar medium. The methodology by Hupkes et al. (2020) primarily evaluates models via specific train and test splits on PCFG SET. `DecompSR`, however, is designed for the prevalent few-shot or zero-shot evaluation paradigm of contemporary LLMs (and reasoning-based LLMs, or 'Large Reasoning Models'), with an aim to diagnose their inherent compositional capabilities acquired from pre-training rather than from specific benchmark fine-tuning.

## 3 The `DecompSR` Dataset

The preceding discussions highlight the critical need for evaluation methods that can dissect the **systematic and compositional reasoning** of LLMs beyond surface-level accuracy. Our proposal is a novel dataset and generation framework which is specifically engineered to facilitate a fine-grained evaluation of compositional spatial reasoning by systematically manipulating the core components of natural language problem instances. While drawing on foundational concepts from (Shi et al., 2022; Li et al., 2024), `DecompSR` implements key design advancements for a controlled, decomposed analysis in line with the compositional facets grounded in Hupkes et al. (2020).

Every `DecompSR` instance is a triple: $\langle s, q, a \rangle$ where, $s$ is a **story**: a sequence of natural language sentences each describing the spatial relation between a pair of adjacent entities (nodes) on an underlying 2-dimensional grid. The construction of $s$ is the primary target for our interventions. $q$ is a **question**, also in natural language, which asks for the spatial relation between two specific entities. These entities are mentioned in $s$ but may not be textually adjacent (i.e. in the same sentence), requiring multi-step inference from the information in $s$. $a$ is the **answer**, a single term representing one of eight primary directions (top, bottom, left, right and the intermediate directions), is the correct spatial relation derived from $s$ in response to $q$. We consider top, bottom, left and right the correct answer if the two points lie on the same horizontal or vertical axis, and the intermediate directions correct if the second queried point lies in the corresponding quadrant with respect to the first one (i.e. if $B$ is in the upper-left quadrant with respect to $A$ then the answer to the query "*where is B with respect to A?*" is $a =$*upper-left*). The framework and the process is presented in Figure 1. Our framework allows for systematically varying elements of the $\langle s, q, a \rangle$ triple through several precisely controlled parameters, each designed to target different aspects of reasoning:

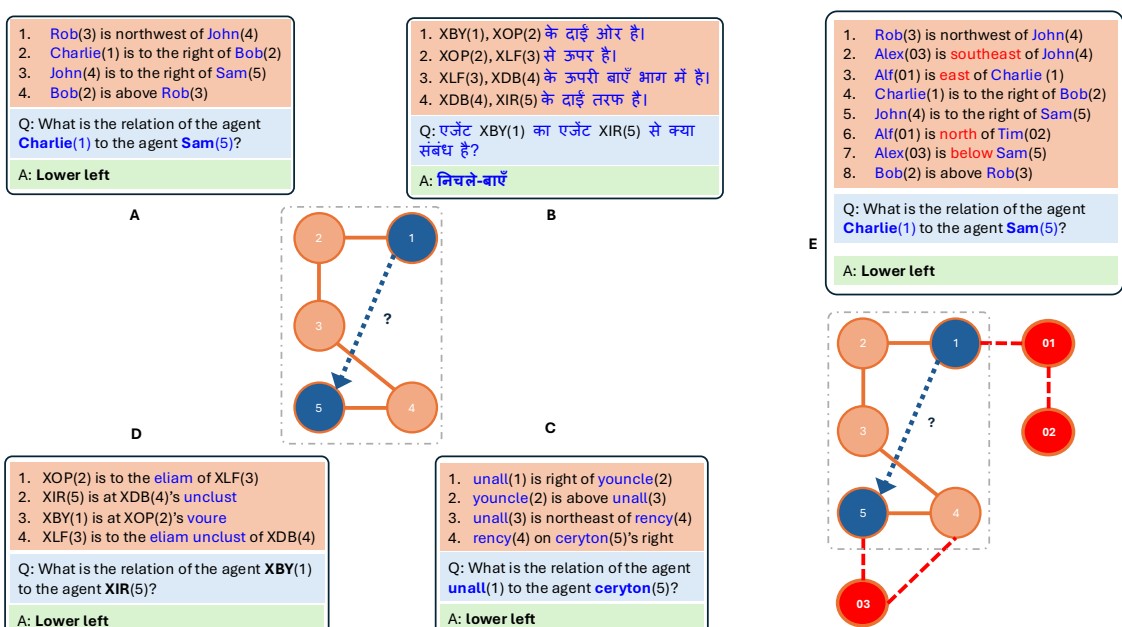

Figure 1: Example of a clean DecompSR data point with $k = 4$. In instance A, we see it instantiated in English, with a shuffled story and male anglophone node names. In instance B, we see the data in Hindi without shuffling the story, using symbolic node names. In C, we use nonsense node names, and in D we have nonsense directions. We have included the numbers (1-5) to help the reader understand the order of the story generation, however in practice these are not explicitly mentioned. On the right we have, in E, an instance of the same core data point (in the grey dashed box) with added noise (nodes 01,02 and 03). The network in the centre is for expository purposes in this paper and is not given to the tested model directly.

**Reasoning Depth ($k$):** This parameter directly controls the complexity of the story $s$ by defining the minimal number of explicit relational sentences within $s$ that must be chained together to correctly determine the answer $a$ for a given question $q$. We have experimented with $k = 1, \ldots, 10, 20, 50, 100$, but our framework supports arbitrary values for $k$.

The variation of $k$ allows assessing a model's **productivity**, i.e., its ability to generalise learned inferential rules to instances of greater complexity or length. Given a few examples (few-shot) in the context of $\langle s, q, a \rangle$ triples with certain $k$ values, we test if a model can accurately derive $a$ from a more complex $s$ (with a larger $k$) for a given $q$. This measures the model's capacity for sustained, chained reasoning as the number of intermediate relational steps in $s$ increases. Our careful construction of $s$ in `DecompSR` ensures $k$ represents the true minimal path length, and thus the minimal number of hops required to answer $q$.

**Language Variation**: This alters the linguistic expression of the relational statements within the story $s$ (and potentially the question $q$). `DecompSR` can generate $s$ and $q$ in multiple natural languages (English, Swedish, Hindi currently benchmarked) or a synthetic English variant where specific directional words within $s$ are replaced by randomly generated nonce words. This directly impacts how the information in $s$ is presented, affecting the derivation of $a$.

Here, the use of nonce words for directions in $s$ primarily tests **systematicity**. After familiarisation with these novel terms in the in-context learning (ICL) example $\langle s, q, a \rangle$ triples, we evaluate if the model can correctly interpret and use these nonce words within new $s$ instances to deduce $a$ for $q$. This assesses if models can treat new symbols in $s$ as systematic replacements for known relational primitives, drawing on concepts of familiarisation (refer to Davidson et al., 2024, interalia).

The availability of different natural languages for $s$ and $q$ allows for testing linguistic **substitutivity**: assessing if the model's ability to derive $a$ is robust to comprehensive changes in linguistic surface form when expressing the same underlying spatial relations in $s$.

**Entity Representation (Node Names)**: This modifies the labels used for entities within both the story $s$ and the question $q$. Options for these names in $s$ and $q$ include symbolic labels, common anglophone first names[1], British city names[2], or randomly generated nonce words[3]. The choice of entity names in $s$ and $q$ allows us to test if the derivation of $a$ is sensitive to such surface changes.

This parameter allows us to assess **substitutivity**. We test if the model's accuracy in deriving $a$ from $s$ and $q$ remains consistent when entity names within $s$ and $q$ are changed (e.g., from symbolic to human names or nonce words), while the underlying relational structure needed to determine $a$ is preserved. Consistent performance indicates robust substitutivity concerning entity types.

**Presence of Distractors (Noise):** This involves augmenting the story $s$ with additional, irrelevant relational sentences that are not on the minimal reasoning path from $s$ required to answer $q$ and derive $a$.

The presence of noise tests a critical aspect of effective reasoning: the ability to *identify and use* only relevant parts of $s$ to derive $a$. The presence of noise also tests **overgeneralisation** when paired with ICL examples which are noise-free. Failures here indicate that models' applications of composition rules is easily disrupted by extraneous information, challenging systematic focus and showing that it generalises the composition of noise-free stories in inappropriate settings.

**Information Order (Story Order):** This controls the sequence of the relational sentences within the story $s$. The sentences in $s$ can be ordered, or fully or partially shuffled. Such manipulation of $s$ probes how models process information to answer $q$ and find $a$ when the input structure varies.

This manipulation of $s$ primarily probes the **overgeneralisation** of order-dependent heuristics. If a model shown ICL examples $\langle s, q, a \rangle$ where $s$ is always ordered then fails to derive $a$ when $s$ is shuffled, it suggests the model overgeneralised an order-based strategy rather than forming an order-invariant representation of the spatial scene in $s$ to answer $q$. This also assesses the robustness of the model's internal 'mental model' construction from $s$.

**On $\langle s, q, a \rangle$ generation:**  The generation of each $\langle s, q, a \rangle$ triple begins by performing a random walk of $k$ steps on a 2-dimensional grid, ensuring no node is revisited within a single walk. This guarantees that the $k$-hop reasoning depth specified for $s$ is exactly equal to the minimal inferential steps needed to link the entities in $q$ and obtain $a$, a crucial improvement over earlier datasets (Shi et al., 2022; Li et al., 2024) where loops could obscure true reasoning depth. Each sentence in the story $s$ is then constructed from this path using diverse natural language templates, where for a given direction there are roughly 20 different template sentences of which one is randomly chosen and filled in with the given node names. An illustration of how these parameters affect the resulting $\langle s, q, a \rangle$ instances is provided in Figure 1.

**On correctness:**  A significant advantage of `DecompSR`'s procedural generation is its inherent correctness and verifiability. Unlike many large-scale benchmarks curated from diverse sources, such as MMLU (Hendrycks et al., 2021a) or other web-derived datasets, which can contain manual annotation errors or ambiguities (Northcutt et al., 2021; McIntosh et al., 2024), every $\langle s, q, a \rangle$ triple in `DecompSR` is correct by construction due to its algorithmic origin. To demonstrate this and provide a reliable upper bound on performance, we employ a purpose-built symbolic solver. This solver features an oracle component that deterministically translates the natural language story $s$ and question $q$ into Answer Set Programs (ASPs). ASP is a declarative logic based programming paradigm based on stable model semantics, well-suited for knowledge representation and logical reasoning (Gelfond & Lifschitz, 1991; Niemelä, 1999). The ASP facts derived from the oracle's translation of $s$ and $q$ are then appended with a predefined ASP knowledge module for spatial reasoning (see Appendix B). The combined program is subsequently processed by the ASP solver `clingo`[4] (Gebser et al., 2011) to deduce the answer $a$. This oracle-based ASP approach effectively reverse-engineers the `DecompSR` generation process, confirming that each problem instance is logically sound and has a unique, deducible answer. This inherent verifiability ensures that any observed model failures are attributable to the model's reasoning capabilities rather than imperfections in the dataset itself.

---

[1] https://raw.githubusercontent.com/facebookresearch/clutrr/refs/heads/develop/clutrr/names.csv
[2] https://geoportal.statistics.gov.uk/datasets/208d9884575647c29f0dd5a1184e711a/about
[3] Ensured to be distinct from the nonce words used for linguistic variation in direction names for probing substitutivity.
[4] https://github.com/potassco/clingo version 5.8.0.

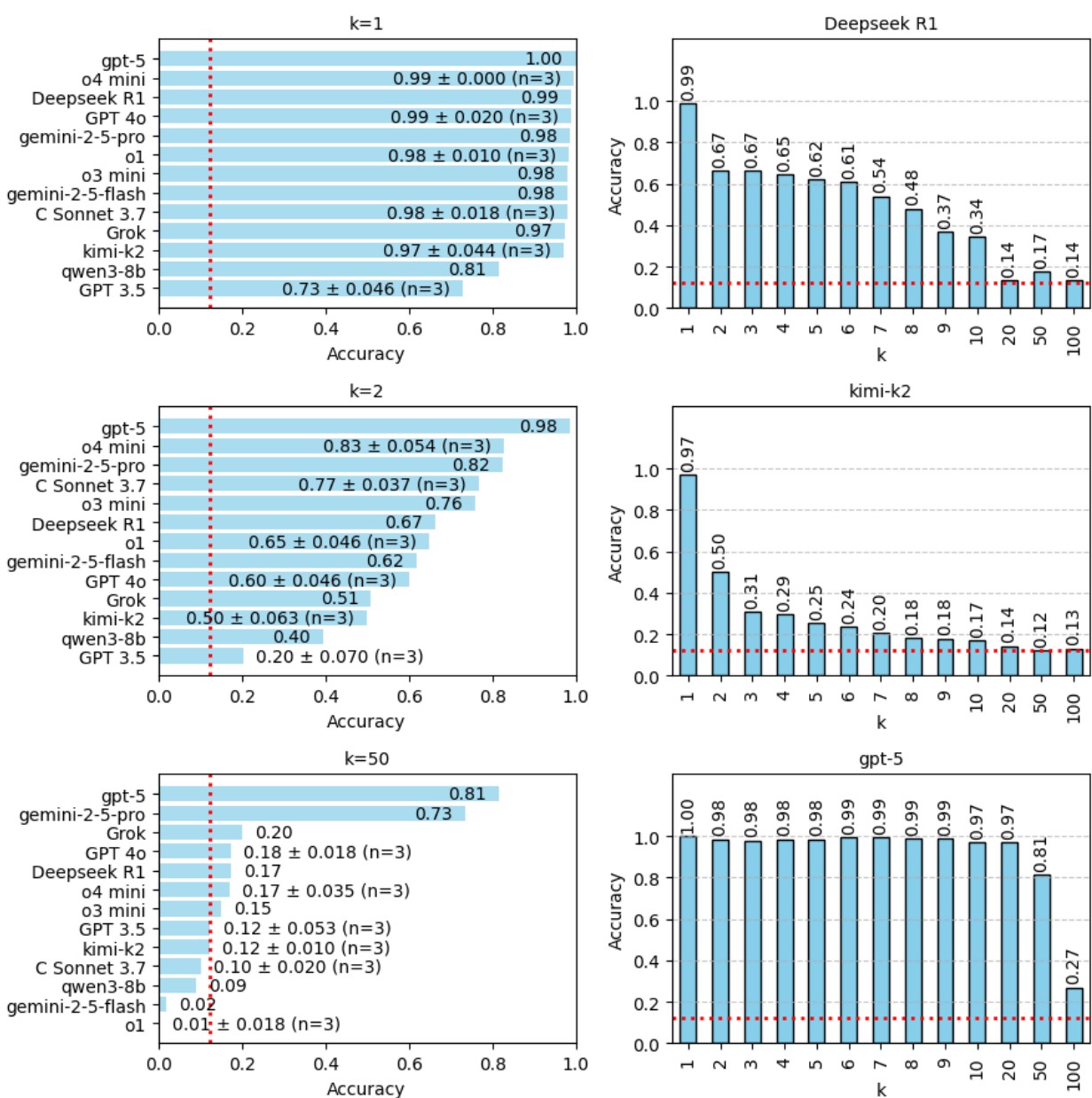

Figure 2: Productivity experiment results. Figures on the left compare model accuracy for $k = 1, 2$ and 50 hop questions. Figures on the right compare the best performing model (GPT-5), with the best open weights models (DeepSeek-R1 and Kimi-K2) for all values of $k$. The red dotted line is the guess rate. The $n$ values shown refer to the number of repetitions, which — where budget allowed — we took to be $n = 3$.

## 4 Benchmarking Experiments

We conduct an experiment for each of the aspects mentioned in Section 3 to demonstrate how this dataset can be used to probe the properties of compositionality. We use a representative selection of contemporary language models detailed in Appendix C. Prompts for all experiments are found in Appendix A.

For the benchmarking experiments, we use DecompSR $200^5$ but we also generated a large version of the dataset (totalling 100,000 default entries per value of $k$ for $k = 1, \ldots, 10, 20, 50, 100$ as well as clean, noisy,

---

[5]Available at https://dataverse.harvard.edu/dataset.xhtml?persistentId=doi:10.7910/DVN/NWDUNY

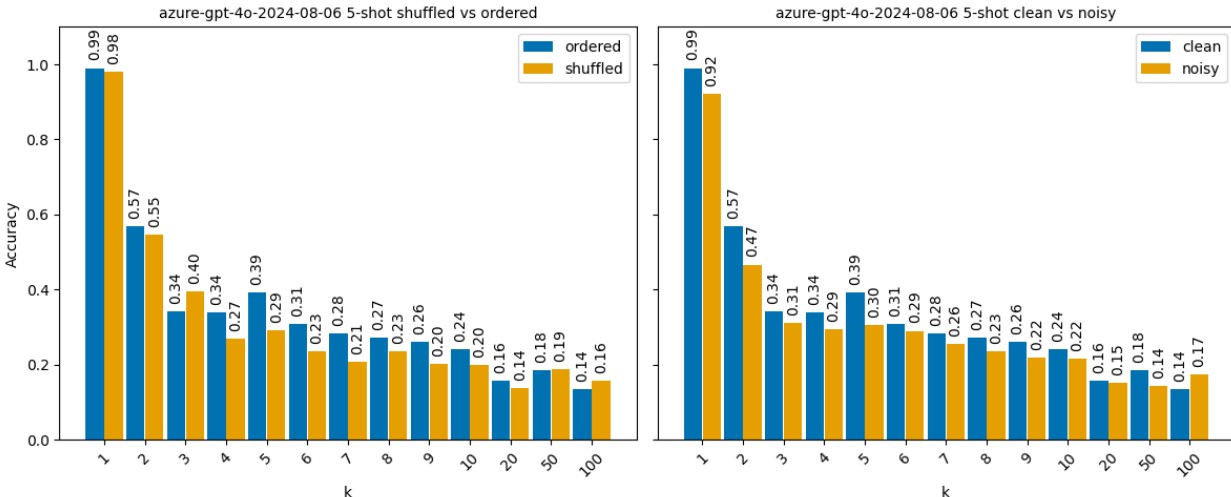

Figure 3: Overgeneralisation experiment for GPT-4o Shuffling the steps reduces accuracy, introducing noise reduces accuracy. Note that for $k = 1$ shuffling has no effect, so the difference here is simply statistical variation.

shuffled and ordered versions of each data point so 5,200,000 entries in total). We also published the code which generated these data in the first place at `https://github.com/navdeepkjohal/DecompSR`.

For the following experiments, we independently vary each of the generating parameters introduced so far. To avoid repetition in defining each experiment, we use the following parameter values as default data in the prompt and baseline, namely: clean, shuffled stories in English with symbolic node names. The standard prompt used (see Appendix A.1) has example stories of lengths $k = 1, 3, 5, 7, 10$, and the baseline experiments are run on $k = 1, \ldots, 10, 20, 50, 100$, unless otherwise stated. The default LLM model for evaluation is GPT-4o, as it served optimally with respect to compute, latency and time constraints. Where budget allowed, we performed 3 repetitions of these experiments and report the mean accuracies and standard deviation.

## 4.1 Productivity Experiment

We test several models' productivity by showing them the 5-shot ICL prompt (Appendix A.1) and then test them on 200 default `DecompSR` examples for each value of $k$ for $k = 1, 2, \ldots, 10, 20, 50, 100$. It is the large range of $k$ which is designed to stress test the productive ability of the model.

The results in Figure 2 demonstrate a general trend across models, that as the number of hops, $k$, increases the accuracy approaches the guess rate (see Appendix D). This suggests that models do not achieve significant productivity. Interestingly, the trend is consistent between reasoning models and LLMs. Among reasoning models, we see stark differences in the accuracy degradation, where all models but GPT-5 and Gemini-2.5-pro are near the guess rate for $k = 10$. For $k = 50$ we see a significant drop in accuracy for GPT-5 and Gemini-2.5-pro, and for $k = 100$ even these models are approaching the guess rate. Further, we provide examples of correct and incorrect predictions made during productivity experiments in Appendix J.

We also conducted the experiment with a 0-shot prompt for GPT-4o as a productivity stress-test (see Table 7 in Appendix D). Accuracy was worse than with 5-shot prompting but showed similar reduction with increasing $k$. Note the code used to generate `DecompSR` allows stories of arbitrary length to be generated. This experiment indicates that conventional models start degrading for low values of $k$, but should one wish to, for example, develop a model particularly apt at productivity, the `DecompSR` dataset allows one to generate stories for arbitrary $k$.

## 4.2 Systematicity Experiment

To evaluate models' systematicity, we test whether the model can capture how to use nonsense words which it has been familiarised with using a variant of the 5-shot ICL prompt (Appendix A.3). This is a copy of

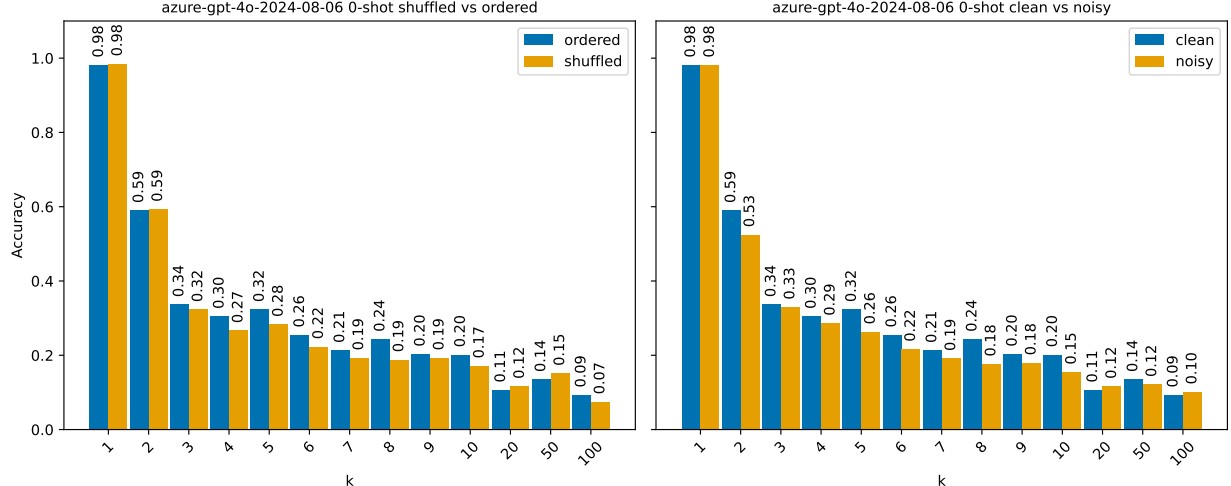

Figure 4: Overgeneralisation experiment for GPT-4o 0-shot. As with the 5-shot experiment, shuffling the steps reduces accuracy, introducing noise reduces accuracy. Note that for k=1 shuffling has no effect.

| $k$ | GPT-4o | | o4-mini | |
|---|---|---|---|---|
| | **English** | **Nonce** | **English** | **Nonce** |
| 1 | $0.99 \pm 0.01$ | $0.37 \pm 0.01$ | $0.99 \pm 0.00$ | $0.51$ |
| 2 | $0.60 \pm 0.01$ | $0.14 \pm 0.01$ | $0.83 \pm 0.02$ | $0.38$ |
| 5 | $0.29 \pm 0.03$ | $0.14 \pm 0.01$ | $0.65 \pm 0.01$ | $0.34$ |
| 10 | $0.21 \pm 0.03$ | $0.10 \pm 0.02$ | $0.52 \pm 0.03$ | $0.26$ |

Table 1: Mean accuracy and standard deviation for English vs. nonsense language in the systematicity experiment run on GPT-4o and o4-mini.

the original prompt where each example has been prepended with a nonsense version of the same prompt, followed by "*This is equivalent to the story*", thus familiarising the model to the nonsense words without giving direct translations. Care has been taken to ensure the entire nonsense vocabulary has been included in the familiarising prompt, see Appendix A.3 for the full familiarisation prompt.

The results of the experiment are presented in Table 1 where we see a drastic difference between the baseline English results and the nonsense results. Although, for $k = 1$, GPT 4o demonstrates some ability for systematic composition, it immediately collapses to the guess rate for $k \geq 2$, showing that it cannot systematically understand and compose. o4-mini has a stronger accuracy than GPT 4o and degrades more slowly but is

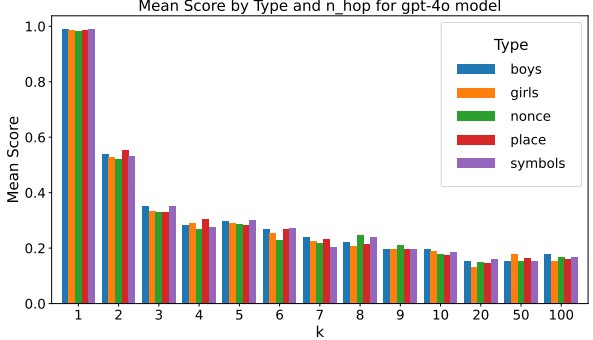

Figure 5: Substitutivity experiment results for GPT-4o model for 5-shot ICL learning.

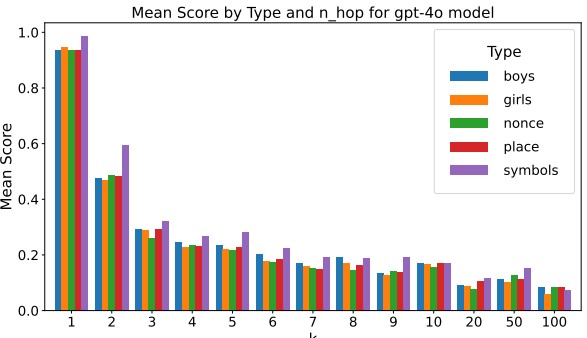

Figure 6: Substitutivity experiment results for GPT-4o model for 0-shot ICL learning.

still approaching the guess rate for $k = 10$. We include the results for all values of $k$ in Table 9 in Appendix E.

### 4.3 Overgeneralisation Experiment

We test overgeneralisation in two ways, first by using a 5-shot ICL prompt for $k = 1, 3, 5, 7, 10$ where the stories are *ordered* (all other properties set to default) and then test the model on `DecompSR` data which have shuffled stories. Note that for small values of $k$, a randomly shuffled story is more likely to be equivalent to the ordered story, and so one would expect a small difference in accuracy between shuffled and ordered data for small values of $k$. The baseline (blue bars in Figure 3) is default `DecompSR` data but with ordered stories. The second method of testing overgeneralisation uses the default 5-shot prompt and tests on *noisy* `DecompSR` data. Here, the baseline is the default `DecompSR` data.

The results for these experiments in Figure 3 (right), show that the accuracy is worse with shuffled data than ordered data and worse with noisy data than clean data. The `DecompSR` dataset can facilitate future experiments investigating the order of reasoning steps and variations in how stories are presented.

We repeated the 5-shot overgeneralisation experiment using 0-shot prompting and results were similar to 5-shot (see Figure 4) but with slightly lower accuracies on average and the slight variation in accuracy in both experiments due to stochasticity of the model. For further analysis of the overgeneralisation experimental results, we present in Appendix H the full results for the 5-shot experiment with o4-mini and GPT-4o in Table 12. We also provide the full 0-shot results for GPT-4o, Llama-3-70B and o1 models in Table 13 in Appendix H.

### 4.4 Substitutivity Experiment

To test substitutivity, we vary the node names of the default `DecompSR` entries. Concretely, this manifests in four sets of experiments one for each set of node names available in `DecompSR`, that is: symbolic (default), anglophone male names, anglophone female names, place names and nonsense names (Appendix C). For each choice of node names, we test the models on `DecompSR` entries with $k = 1, \ldots, 10, 20, 50, 100$ where each of them have matching ICL-prompts but only for $k = 1, 3, 5, 7, 10$. Concretely, we conduct five sets of experiments each corresponding to distinct naming scheme for symbolic nodes (default) present in `DecompSR` dataset. We replace symbolic names with symbolic anglophone male names, anglophone female names, nonsense names and city names.

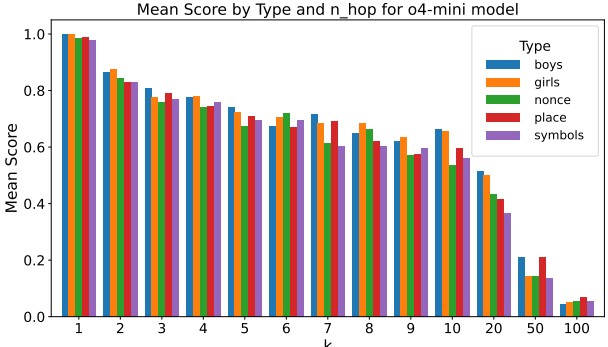

Figure 7: Substitutivity experiment results for o4-mini model for 5-shot ICL learning.

The experiments are performed by using GPT-4o and o4-mini models for $k = 1, 2, \ldots, 10, 20, 50, 100$. To ensure reproducibility, each experiment with GPT-4o is repeated three times whereas the experiments with o4-mini are conducted once because of the

| $k$ | GPT-4o | | | o4-mini | | |
|---|---|---|---|---|---|---|
| | **English** | **Hindi** | **Swedish** | **English** | **Hindi** | **Swedish** |
| 1 | $0.99 \pm 0.01$ | $0.96 \pm 0.01$ | $0.82 \pm 0.01$ | $0.99 \pm 0.00$ | 0.95 | 0.30 |
| 2 | $0.60 \pm 0.01$ | $0.47 \pm 0.02$ | $0.43 \pm 0.01$ | $0.83 \pm 0.02$ | 0.73 | 0.23 |
| 5 | $0.29 \pm 0.03$ | $0.22 \pm 0.01$ | $0.26 \pm 0.01$ | $0.65 \pm 0.01$ | 0.63 | 0.09 |
| 10 | $0.21 \pm 0.03$ | $0.18 \pm 0.01$ | $0.20 \pm 0.00$ | $0.52 \pm 0.03$ | 0.48 | 0.07 |

Table 2: Mean accuracy ($\pm$ standard deviation) across languages (English, Hindi, Swedish) for `GPT-4o` and `o4-mini` under 5-shot ICL.

resource limitations. The results obtained by using 0-shot ICL and 5-shot ICL with GPT-4o model are shown in Figures 5 and 6 respectively while the results for 5-shot ICL with o4-mini model are shown in Figure 7. As apparent from the figures, both GPT-4o and o4-mini are relatively insensitive to the choice of node names allowing us to conclude that these models exhibit strong capacity for substitutivity.

As a secondary substitutivity experiment we consider a multilingual experiment, providing an interesting supplementary study on how linguistic variation affects reasoning performance. Generating `DecompSR` instances in multiple languages amounts to translating the natural language template, which we have done in Hindi and Swedish by first machine translating the existing English template using Google Translate[6] , DeepSeek[7], ChatGPT[8] and let native speakers correct and choose the best translations from the different models. The experiment was performed on 200 default `DecompSR` entries (i.e. ordered, no-noise, symbolic names) per value of $k$ for $k = 1, 2, 5, 10$. The accuracy for the Swedish experiment is significantly worse than Hindi and English for low $k$ but as $k$ increases, all accuracies trend towards the guess rate (Table 2).We present the results of `o4-mini`, along with more detailed results for `GPT-4o`, see Table 10 in Appendix F.

We performed an analysis of the difference in token usage across the different languages, with full results presented in Appendix I in Table 14. We found that prompts were tokenised using more tokens in Hindi than Swedish and English (which were comparable). However both Hindi and Swedish used more completion tokens than English experiments, with Swedish requiring more than Hindi. This follows trends in multilingual tokenization like (Petrov et al., 2023).

### 4.5 Evaluating Reasoning Tasks Symbolically

We also use LLMs to translate `DecompSR` stories into Answer Set Programming (ASP) facts, on which we run `clingo` (see the end of Section 3) to reason symbolically. This task tests whether models can consistently abstract natural language into a simple relational format. Here we tested models on 400 instances of `DecompSR` for each value of $k$ ranging over $1, \dots, 10, 20, 50, 100$ for both clean and noisy instances, resulting in $400 \times 13 \times 2 = 10,400$ datapoints.

Models were prompted to translate the `DecompSR` into ASP using an ICL prompt of a single `DecompSR` instance with $k = 10$. This may be viewed as a 10-shot prompt as each of the 10 examples were single natural language sentences and their ASP counterparts. Moreover there is one query sentence and its ASP-query counterpart (see the prompt in full in Appendix A.6). We chose $k = 10$ as a rough average of $k$ across the experiment and to ensure full coverage of all the possible directions in the ASP-vocabulary.

| Type | $k \to$ Model | 1 | 10 | 20 | 50 | 100 |
|---|---|---|---|---|---|---|
| | GPT-4o | 0.9 | 0.9 | 0.8 | 0.6 | 0.2 |
| | gem-2.5-f | 1.0 | 0.9 | 0.9 | 0.7 | 0.5 |
| | o4-mini | 1.0 | 0.9 | 0.8 | 0.7 | 0.5 |
| LLM+ASP | gpt5.1 | 0.9 | 0.8 | 0.7 | 0.5 | 0.3 |
| | kimi-k2 | 0.4 | 0.2 | 0.3 | 0.1 | 0.1 |
| | DeepSeek | 0.8 | 0.8 | 0.7 | 0.6 | 0.4 |
| | GPT OSS | 0.8 | 0.8 | 0.7 | 0.5 | 0.3 |
| Oracle+ASP | | 1.0 | 1.0 | 1.0 | 1.0 | 1.0 |

Table 3: Accuracy by $k$ for GPT-4o, Gemini 2.5 Flash (`gem-2.5-f`), GPT o4-mini, GPT 5.1, kimi-k2, DeepSeek and GPT OSS models, averaged over clean and noisy ASP runs. All models were implemented with default settings (see Appendix C for full detail.)

---

[6] https://translate.google.com/
[7] https://www.deepseek.com/
[8] https://chatgpt.com/

Models were tasked with translating all sentences in the story in one pass, not one-by-one. As the $k = 1$ results show, models are able to translate single `DecompSR` sentences into ASP facts, however for large $k$ (i.e. multiple sentences) the translations start to break down showing brittleness even while translating (Table 3). We present full results in Table 11 in Appendix G

A correctly translated `DecompSR` story would result in a single answer formula in the semantics of the resulting ASP program. However, if the model translates incorrectly, it is possible there are multiple answer formulas present. We consider any case where there are more than one answer formula incorrect which is a more stringent approach. We report the frequency of the different errors in Section 5.1, along with further error analysis on the nature and causes of the errors. The results from the Oracle+ASP experiments verify that the examples are accurate, thus highlighting that it is indeed the LLM which lacks robustness, potentially abstracting necessary information at larger $k$.

## 5 Discussion

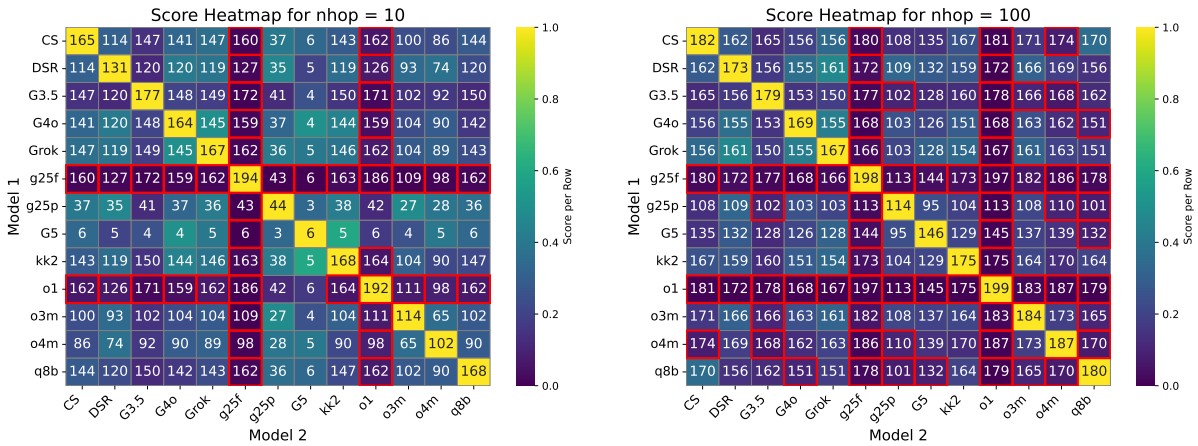

Figure 8: Number of incorrect answers in common across tested models for $k = 10$ and $k = 100$ of the productivity experiment in Section 4.1. The numbers are the number of problems both models failed and the colouring represents the number of incorrectly answered problems which were answered with the same incorrect answer, with yellow representing high proportion of same incorrect answers, and blue representing different incorrect answers. Red boxes denote when the both models perform below the guess rate. `CS`, `DSR`,`G3.5`, `G4o`, `g25f`, `g25p`, `G5`, `kk2`, `o3m`, `o4m`, `q8b` represent Claude Sonnet, Deepseek R1, GPT 3.5, GPT-4o, gemini-2-5-flash, gemini-2-5-pro, gpt-5, kimi-k2, o3 mini, o4-mini, qwen3-8b respectively.

### 5.1 Failure Modes

We analysed the types of failures of the models in two key experiments: the productivity experiment in Section 4.1 and the symbolic translation experiment in Section 4.5.

Across the benchmarking (productivity) experiments we notice that as the reasoning depth increases, language models tend to guess randomly while reasoning models start abstaining. This was observed by correlating the incorrect model predictions across the different models to see if there were any patterns in the failures; as seen in Figure 8 there is little agreement in predictions across models. We note also that the oracle+ASP results from the symbolic translation experiments guarantee that there is indeed sufficient information to solve the `DecompSR` question, rendering abstentions as incorrect answers.

In the symbolic translation experiment in Section 4.5, we highlight that mistakes are purely a result of mistranslation, rather than lacking compositionality since the oracle+ASP experiment results show that there is no issue with the ASP-rules used to reason over `DecompSR` problems. Mistranslating here means that the model either makes syntactically incorrect translations (i.e. not following ASP-syntax), semantically

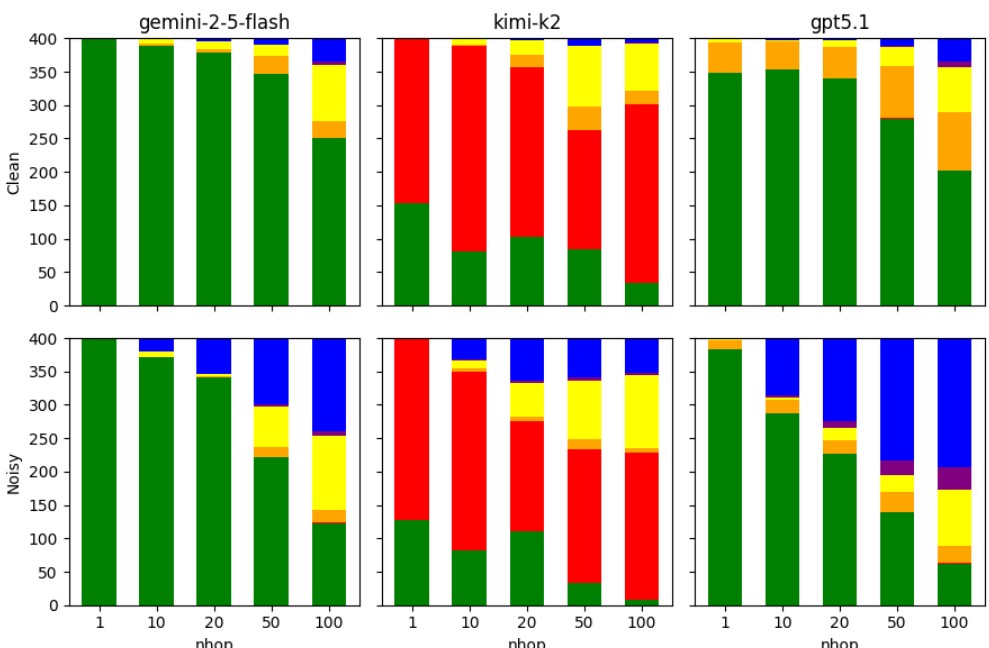

Figure 9: Syntactic and semantic errors per model per $k$. The colour-coding shows green : correct, red : syntactic error, orange : single incorrect answer, yellow : no answer, purple : multiple incorrect answers, blue : multiple answers of which at least one is correct.

incorrect translations (like "*A is to the left of B.*" being translated to `left(B,A).`), omits translations or hallucinates ASP-facts or queries.

We present a breakdown of syntactic and semantic errors for key models in Figure 9 (and the full results in Figure 11 in Appendix K). We see that as $k$ increases, the overall number of errors increases and that for larger models the errors tend to be semantic, whereas for smaller models the errors are more often syntactic.

Semantic errors take several forms: the simplest is a single incorrect answer (e.g. the semantics contains a single answer-formula `answer(left)`, when the correct answer is `right`), these are represented by orange bars; then there is the case where there is no answer formula despite the program compiling correctly, represented by yellow bars. There is also the case where the resulting program yields multiple answer formulas, of which none are correct (e.g. `answer(left), answer(below)`) represented by purple bars, or at least one is correct (e.g. `answer(left), answer(right)` when the answer is `right`) represented by blue bars.

The syntactic errors themselves are mostly minor flaws, such as missing end-of-clause symbols required by ASP-syntax, but nevertheless cause compilation errors when running `clingo`. The presence of noise lowers accuracy by introducing facts that lead to multiple answers being generated, of which one is the correct answer (compare the blue bars between clean and noisy runs in Figure 9). This further evidences that models struggle with productivity, since in this experiment models are only tasked with translating, and the presence of noise simply increases the number of sentences to be translated. Hence for any $k$, one can predict that for noisy instances the accuracy will be lower than for cleaner ones.

Semantic errors are either caused by hallucinated ASP-facts leading to incorrect answers, superfluous or missing ASP-queries leading to multiple or no answers, respectively. We analysed these occurrences and found that models generating superfluous queries was rare (the highest rate was 0.2% for the model GPT-OSS-20B.) Instead it was hallucinated facts (either superfluous or incorrect) which contributed to there being so many occurrences of multiple answers, see Figure 10 for the counts and Figure 12 in Appendix K

for full results. Note that we are guaranteed errors when models generate too few facts for clean stories due to the nature of `DecompSR` being correct and minimal by construction. That is, for clean stories we have the necessary and sufficient information to answer the query, so missing any fact will guarantee it is impossible to answer the query.

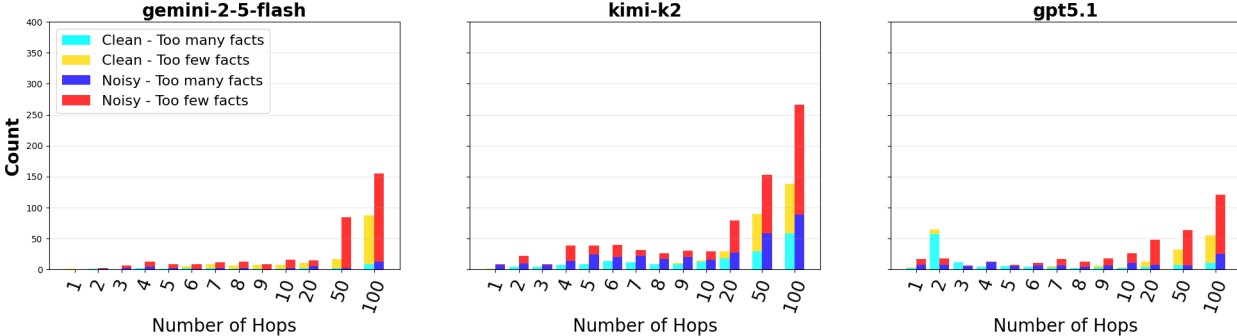

Figure 10: Counts of ASP-facts, colour coded by noise in blues vs reds for too few vs too many facts generated, and in dark vs light shades indicating noisy and clean data points.

We note some trends across the type of models used, where in general the size of model will lead to considerably fewer syntactic errors, and broadly speaking reasoning models outperformed language models in terms of raw accuracy. The three models analysed in Figure 9 represented the state of the art at the time of the experiments, in terms of reasoning models (Gemini-2.5-flash), open models (Kimi-K2) and large language models (GPT-5.1), but we include the same analysis for all tested models in Figure 11 in the Appendix K.

## 5.2 Future Directions

`DecompSR` achieves the goal of benchmarking the decomposed compositional abilities of language models in 2D spatial reasoning, which may lead to several directions for further datasets for spatial reasoning which involve further grounding, modality or complexity.

As for grounding, a next step would be to situate `DecompSR` stories in a natural language context so that stories do not occur in an 2D grid but say a map, or a description of a landscape. Note that `DecompSR` as is provides a clear, simple task which we have demonstrated already challenges models to reason productively and systematically. Further grounding `DecompSR` in a more natural context, does not immediately add to the analysis of compositionality, but instead perhaps the ability of models to abstract and form mental-models. Developing `DecompSR` in this direction would thus become an interesting pursuit for developing cognitive science benchmarks regarding mental models for spatial reasoning.

On grounding, there may be further opportunities to ground `DecompSR` by moving to 3D data. Although working in 2D already answers the questions we posed regarding compositionality, the added complexities of 3D spatial reasoning may provide an even richer benchmark if `DecompSR` as it stands is ever considered 'solved'.

In spatial reasoning more holistically, vision is a crucial modality which may be interesting to integrate into `DecompSR`. We have left this as a future avenue since, as it stands, the understanding of compositionality in vision is less granular than that of text, as mentioned explicitly in Vegner et al. (2025). It is not clear how one would, for instance, analyse a vision model's systematicity in a spatial reasoning context at least in connection to `DecompSR`. Such an integration of textual and visual spatial reasoning data would be interesting in the analysis of reasoning capacities of vision language models (VLMs).

Another avenue to pursue is to increase the linguistic and logical or mathematical complexity of `DecompSR`. Regarding linguistic complexity, we still intend to maintain a templating approach so as to guarantee the correctness of the natural language data, but where one adds more complex grammatical, semantic or even pragmatic features. This might mean integrating pronominal references like *A* and *it* coreferring in "*below*

*B lies A which has C to the left of it*" or broadening the templating to include patterns like mentioning the relations between three or more nodes rather than only two. Presently, models would likely perform worse on such a dataset, which would lead to questions of if models failed in terms of compositional reasoning or in terms of natural language understanding. The simplicity of `DecompSR` isolates failures of the models to failures of compositional reasoning which the benchmarking in Section 4 shows happens across all models.

As for logical or mathematical complexity, a simple addition would be to include distances between nodes, and letting this distance vary. Such a version of `DecompSR` would require a combination of 2D spatial reasoning as in the current instance of the dataset, as well as arithmetic ability. The research questions asked in this paper however regard compositional reasoning ability, and we argue that in its current form `DecompSR` isolates this neatly and adding arithmetic would blur the understanding of compositional reasoning ability. Instead such an version of `DecompSR` would be useful for benchmarking more complex spatial reasoning stories, departing from the purer evaluation of spatial compositional abilities.

## 6 Conclusions

We have presented `DecompSR`, a benchmarking framework with over 5 Million samples for the decomposed analysis of multi-hop compositional spatial reasoning. `DecompSR`'s core contribution lies in its methodology: the systematic generation of natural language task instances ($\langle s, q, a \rangle$ triples) where parameters are precisely controlled to probe distinct compositional abilities —productivity, systematicity, substitutivity, and overgeneralisation—as inspired by the framework of Hupkes et al. (2020). A key design feature is its inherent verifiability through procedural generation and an accompanying ASP-based oracle solver, ensuring dataset correctness and allowing for clear attribution of performance to model capabilities. Our benchmarking experiments on a range of contemporary LLMs using `DecompSR` revealed important insights. While models demonstrated a degree of robustness to linguistic variation (substitutivity of relational phrases in different languages and tolerance for varied entity names), they exhibited significant limitations in productivity, with performance degrading substantially as the number of reasoning hops ($k$) increased. Our analysis on failure modes has shown how, when failing to answer the question randomly guess rather than consistently guess the same wrong answer, giving insights into the chain of thought (or lack thereof) among tested models. The symbolic translation experiment demonstrates that a simple tool-call and prompt is generally speaking a better method for solving `DecompSR` problems.

## 7 Broader Impact

We raise the potential harmful impacts of `DecompSR` arising from training or fine-tuning LLMs on `DecompSR` being deployed in spatial reasoning application in security-critical domains like robotics or navigation. Any errors present in a benchmark may have direct and indirect impacts in applications. To mitigate the presence of errors we have gone to great lengths to ensure `DecompSR` is correct by construction, and have verified it to be correct within our framework. We have developed `DecompSR` as a dataset for evaluating, even stress-testing compositional abilities — we do not foresee this having any military applications, but note here that improvements in automated reasoning come with the potential for misuse and should be carefully conducted.

Although the generation of `DecompSR` is computationally efficient, the benchmarking experiments presented in Section 4 required extensive use of LLMs, incurring substantial financial costs and associated environmental impacts, including $CO_2$ emissions and water consumption. We acknowledge this here and encourage further use of `DecompSR` to be done at smaller scales, either with smaller models or fewer data points.

Since `DecompSR` is produced using a generative procedure, we foresee that should there be leakage of published `DecompSR` data into LLMs through training or fine-tuning, one can readily generate fresh `DecompSR` data to ensure models cannot use any learned shortcuts. This improves the longevity and utility of `DecompSR` for the reasoning research community.

## 8 Acknowledgments

This work was supported in part by the Alan Turing Institute under Fundamental Research (Project No. PP00029), the Economic and Social Research Council (ESRC) under grant ES/W003473/1, by the EPSRC under grant EP/Z003512/1, and by the Special Funds of Tongji University for the "Sino-German Cooperation 2.0 Strategy". We thank Microsoft Research – Accelerating Foundation Models Research program, for the provision of Azure resources to access OpenAI models, without which many of the experiments in this paper would not have been possible.

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

# A LLM Prompts

## A.1 5-shot Default Prompt

```
1    {{"id": "{line['ID']}",
2     "messages": [
3      {{"role": "user","content":"Given a story about spatial relations among objects, answer
            the relation between two queried objects. Possible relations are: above, below, left,
             right, upper-left, upper-right, lower-left, and lower-right. If a sentence in the
            story is describing clock-wise information, then 12 denotes above, 1 and 2 denote
            upper-right, 3 denotes right, 4 and 5 denote lower-right, 6 denotes below, 7 and 8
            denote lower-left, 9 denote left, 10 and 11 denote upper-left. If the sentence is
            describing cardinal directions, then north denotes above, east denotes right, south
            denotes below, and west denotes left.\\nAnswer the question and provide the final
            answer in the form: '### Answer:'\\n\\nStory:\\n1 XU is to the right and below XJX at
             an angle of about 45 degrees.\\n\\nWhat is the relation of the agent XU to the agent
             XJX?"}},
4
5      {{"role": "assistant", "content": "### Answer: lower-right"}},
6
7      {{"role": "user", "content": "Story:\\n1 XEX is to the bottom right of XEM.\\n2 XFR is
            positioned up and to the right of XEM.\\n3 XEX is to the left of XJM with a small gap
            between them.\\n\\nWhat is the relation of the agent XJM to the agent XFR?"}},
8
9      {{"role": "assistant", "content": "### Answer: lower-right"}},
10
11
12     {{"role": "user", "content": "Story:\\n1 XAV is positioned right to XH.\\n2 XH is on the
            right and XDC is on the left.\\n3 XAE and XJT are vertical and XAE is below XJT.\\n4
            XDC presents over XJT.\\n5 XEG is sitting at the 6:00 position to XAE.\\n\\nWhat is
            the relation of the agent XAV to the agent XEG?"}},
13
14     {{"role": "assistant", "content": "### Answer: upper-right"}},
15
16
17     {{"role": "user", "content": "Story:\\n1 The object labeled XBK is positioned to the right
             of the object labeled XGX.\\n2 XDT is over XGX.\\n3 XIC is below XDT and to the left
            of XDT.\\n4 XIC and XBD are in a vertical line with XIC on top.\\n5 XBD is south east
            of XFT.\\n6 If XFT is the center of a clock face, XDV is located between 7 and 8.\\n7
            XDV is positioned below XFY.\\n\\nWhat is the relation of the agent XFY to the agent
            XBK?"}},
18
19     {{"role": "assistant", "content": "### Answer: left"}},
20
21
22     {{"role": "user", "content": "Story:\\n1 XJV and XEJ are horizontal and XJV is to the left
             of XEJ.\\n2 XJV is directly north east of XEX.\\n3 XEX is to the right of XDU
            horizontally.\\n4 XDU and XCF are vertical and XDU is above XCF.\\n5 XQ is on the left
             side and above XCF.\\n6 XQ and XGQ are side by side with XQ to the right and XGQ to
            the left.\\n7 XGQ is over there and XIY is directly above it.\\n8 The object XIY and
            XDV are there. The object XIY is below and slightly to the right of the object XDV.\\
            n9 XCN is at a 45 degree angle to XDV, in the lower lefthand corner.\\n\\nWhat is the
            relation of the agent XCN to the agent XEJ?"}},
23
24     {{"role": "assistant", "content": "### Answer: left"}},
25
26
27     {{"role": "user", "content": "Story:\\n{line['data'].replace('\n', '\\n')}\\n{line['
            question'].replace('\n', '\\n')}"}}]}}
```

## A.2 0-shot Prompt (Productivity Experiment)

```
1    {{"id": "{line["ID"]}",
2     "messages": [
3      {{"role": "user","content":
4       "Given a story about spatial relations among objects, answer the relation between two
             queried objects. Possible relations are: above, below, left, right, upper-left, upper
             -right, lower-left, and lower-right. If a sentence in the story is describing clock-
             wise information, then 12 denotes above, 1 and 2 denote upper-right, 3 denotes right,
              4 and 5 denote lower-right, 6 denotes below, 7 and 8 denote lower-left, 9 denote
             left, 10 and 11 denote upper-left. If the sentence is describing cardinal directions,
              then north denotes above, east denotes right, south denotes below, and west denotes
             left.\\nAnswer the question and provide the final answer in the form: '### Answer:'\\
             n\\nStory:\\n{line["data"].replace("\n", "\\n")}\\n{line["question"].replace("\n",
             "\\n")}"}}]}}
```

## A.3  5-shot Familiarisation Prompt

```
1    {{"id": "{line['ID']}",
2     "messages": [
3      {{"role": "user","content":"Given a story about spatial relations among objects, answer
             the relation between two queried objects. You will be given the directions in an
             artificial language, where the possible relations in English are: above, below, left,
              right, upper-left, upper-right, lower-left, and lower-right.\\nAnswer the question
             and provide the final answer in the form: '### Answer:'\\n\\nStory:\\n1 XU is to the
             absol voure of XJX.\\n\\nWhat is the relation of the agent XU to the agent XJX?\\n\\
             nThis is equivalent to the story\\n1 XU is to the right and below XJX at an angle of
             about 45 degrees.\\n\\nWhat is the relation of the agent XU to the agent XJX?"}},
4
5      {{"role": "assistant", "content": "### Answer: lower-right"}},
6
7      {{"role": "user", "content": "Story:\\n1 XEX is to the meanion writent of XEM.\\n2 XFR is
             to the eliam voure of XEM.\\n3 XEX is at XJM's unclust.\\n\\nWhat is the relation of
             the agent XJM to the agent XFR?\\n\\nThis is equivalent to the story:\\n1 XEX is to
             the bottom right of XEM.\\n2 XFR is positioned up and to the right of XEM.\\n3 XEX is
             to the left of XJM with a small gap between them.\\n\\nWhat is the relation of the
             agent XJM to the agent XFR?"}},
8
9      {{"role": "assistant", "content": "### Answer: lower-right"}},
10
11
12     {{"role": "user", "content": "Story:\\n1 XAV is to the writent of XH.\\n2 XH is at XDC's
             voure.\\n3 XAE is to the meanion of XJT.\\n4 XDC is to the eliam of XJT.\\n5 XEG is at
              XAE's voure.\\n\\nWhat is the relation of the agent XAV to the agent XEG?\\n\\nThis
             is equivalent to the story:\\n1 XAV is positioned right to XH.\\n2 XH is on the right
             and XDC is on the left.\\n3 XAE and XJT are vertical and XAE is below XJT.\\n4 XDC
             presents over XJT.\\n5 XEG is sitting at the 6:00 position to XAE.\\n\\nWhat is the
             relation of the agent XAV to the agent XEG?"}},
13
14     {{"role": "assistant", "content": "### Answer: upper-right"}},
15
16
17     {{"role": "user", "content": "Story:\\n1 XBK is to the writent of XGX.\\n2 XDT is at XGX's
              meanion unclust.\\n3 XIC is to the absol imach of XDT.\\n4 XIC is at XBD's picited.\\
             n5 XBD is to the absol voure of XFT.\\n6 If XFT is at XDV's picited writent.\\n7 XDV
             is to the meanion of XFY.\\n\\nWhat is the relation of the agent XFY to the agent XBK
             ?\\n\\nThis is equivalent to the story:\\n1 The object labeled XBK is positioned to
             the right of the object labeled XGX.\\n2 XDT is over XGX.\\n3 XIC is below XDT and to
             the left of XDT.\\n4 XIC and XBD are in a vertical line with XIC on top.\\n5 XBD is
             south east of XFT.\\n6 If XFT is the center of a clock face, XDV is located between 7
             and 8.\\n7 XDV is positioned below XFY.\\n\\nWhat is the relation of the agent XFY to
             the agent XBK?"}},
18
19     {{"role": "assistant", "content": "### Answer: left"}},
20
21
22     {{"role": "user", "content": "Story:\\n1 XJV is at XEJ's unclust.\\n2 XJV is at XEX's
             picited.\\n3 XEX is to the writent of XDU.\\n4 XDU is to the eliam of XCF.\\n5 XQ is
             to the eliam unclust of XCF.\\n6 XQ is at XGQ's voure.\\n7 XGQ is to the meanion of
             XIY.\\n8 XIY is at XDV's meanion writent.\\n9 XCN is to the absol imach of XDV.\\n\\
```

```
        nWhat is the relation of the agent XCN to the agent XEJ?\\n\\nThis is equivalent to
        the story:\\n1 XJV and XEJ are horizontal and XJV is to the left of XEJ.\\n2 XJV is
        directly north east of XEX.\\n3 XEX is to the right of XDU horizontally.\\n4 XDU and
        XCF are vertical and XDU is above XCF.\\n5 XQ is on the left side and above XCF.\\n6
        XQ and XGQ are side by side with XQ to the right and XGQ to the left.\\n7 XGQ is over
        there and XIY is directly above it.\\n8 The object XIY and XDV are there. The object
        XIY is below and slightly to the right of the object XDV.\\n9 XCN is at a 45 degree
        angle to XDV, in the lower lefthand corner.\\n\\nWhat is the relation of the agent XCN
        to the agent XEJ?"}},
23
24      {{"role": "assistant", "content": "### Answer: left"}},
25
26      {{"role": "user", "content": "Story:\\n{line['data'].replace('\n', '\\n')}\\n{line['
        question'].replace('\n', '\\n')}"}}]}}
```

## A.4   5-shot Hindi Prompt

{"role": "user","content":"वस्तुओं के बीच स्थानिक संबंधों के बारे में एक कहानी दी गई है, दो पूछी गई वस्तुओं के बीच संबंध का उत्तर दीजिए। संभावित संबंध हैं: ऊपर, नीचे, बाएं, दाएँ, ऊपरी–बाएँ, ऊपरी–दाएँ, निचले–बाएँ और निचले–दाएँ। यदि कहानी में कोई वाक्य घड़ी की दिशा में सूचना का वर्णन कर रहा है, तो 12 ऊपर को दर्शाता है, 1 और 2 ऊपरी–दाएँ को दर्शाता है, 3 दाएं को दर्शाता है, 4 और 5 निचले–दाएँ को दर्शाता है, 6 नीचे को दर्शाता है, 7 और 8 निचले–बाएँ को दर्शाता है, 9 बाएं को दर्शाता है, 10 और 11 ऊपरी–बाएँ को दर्शाता है। यदि वाक्य मुख्य दिशाओं का वर्णन कर रहा है, तो उत्तर ऊपर को दर्शाता है, पूर्व दाई ओर को दर्शाता है, दक्षिण नीचे को दर्शाता है, और पश्चिम बाई ओर को दर्शाता है।\n प्रश्न का उत्तर दें और अंतिम उत्तर इस रूप में दें: '###        :'\n\nकहानी:\n1 XU, XJX के दाई ओर और लगभग 45 डिग्री के कोण पर नीचे है।\n\n एजेंट XU का एजेंट XJX से क्या संबंध है?"},

{"role": "assistant", "content": "### उत्तर: निचले–दाएँ"},

{"role": "user", "content": "कहानी:\n1 XEX, XEM के नीचे दाई ओर है।\n2 XFR, XEM के ऊपर और दाई ओर स्थित है।\n3 XEX, XJM के बाई ओर है और उनके बीच थोड़ा अंतर है।\n\n एजेंट XJM का एजेंट XFR से क्या संबंध है?"},

{"role": "assistant", "content": "### उत्तर: निचले–दाएँ"},

{"role": "user", "content": "कहानी:\n1 XAV, XH के दाई तरफ स्थित है।\n2 XH दाई ओर है और XDC बाई ओर है।\n3 XAE और XJT ऊर्ध्वाधर हैं और XAE, XJT के नीचे है।\n4 XDC, XJT के ऊपर मौजूद है।\n5 XEG, XAE की घड़ी में 6 बजे की स्थिति में बैठा है।\n\nएजेंट XAV का एजेंट XEG से क्या संबंध है?"},

{"role": "assistant", "content": "### उत्तर: ऊपरी–दाएँ"},

{"role": "user", "content": "कहानी:\n1 XBK नामक  वस्तु, XGX नामक  वस्तु  के  दाई  ओर  स्थित है।\n2 XDT, XGX के ऊपर है।\n3 XIC, XDT के नीचे और XDT के बाई ओर है।\n4 XIC \verbऔर XBD एक ऊर्ध्वाधर रेखा में हैं, जिसमें XIC ऊपर है।\n5 XBD, XFT के दक्षिण–पूर्व में है।\n6 यदि XFT घड़ी के मुख का केंद्र है, तो XDV 7 और 8 के बीच स्थित है।\n7 XDV, XFY के नीचे स्थित है।\n\nएजेंट XFY का एजेंट XBK से क्या संबंध है?"},

{"role": "assistant", "content": "###     :  "},

{"role": "user", "content": "Story:\n1XJV और XEJ क्षैतिज हैं और XJV, XEJ के बाई ओर है।\n2 XJV, XEX के ठीक उत्तर–पूर्व में है।\n3 XEX क्षैतिज रूप से XDU के दाई ओर है।\n4 XDU और XCF ऊर्ध्वाधर हैं और XDU, XCF के ऊपर है।\n5 XQ, XCF के बाई ओर और ऊपर है।\n6 XQ और XGQ साथ–साथ हैं, जिसमें XQ दाई ओर और XGQ बाई ओर है।\n7 XGQ वहाँ है और XIY उसके सीधे ऊपर है।\n8 वस्तुएँ XIY  और XDV  वहाँ हैं। वस्तु XIY, वस्तु XDV से नीचे और थोड़ा दाई ओर है।\n9 XCN, XDV से 45 डिग्री के कोण पर निचले–बाएँ कोने में है।\n\nएजेंट XCN का एजेंट XEJ से क्या संबंध है?"},

{"role": "assistant", "content": "### उत्तर: बाएं"},

{"role": "user", "content": "Story:\n1 XIC, XAL के ऊपर रखा गया है।\n\nएजेंट XIC का एजेंट XAL से क्या संबंध है?"}

## A.5   5-shot Swedish Prompt

```
1       {{"id": "{line['ID']}",
2        "messages": [
```

```
3        {{"role": "user","content":"Givet en berättelse om rumsliga relationer mellan objekt,
             besvara relationen mellan två objekt som frågas. De möjliga relationerna är: ovan,
             nedan, vänster, höger, ovan-vänster, ovan-höger, nedan-vänster och nedan-höger. Om en
              mening i berättelsen beskriver information som går medsols, betecknar 12 ovan, 1 och
             2 ovan-höger, 3 höger, 4 och 5 nedan-höger, 6 nedan, 7 och 8 nedan-vänster, 9
             vänster, 10 och 11 ovan-vänster. Om meningen beskriver väderstreck, betecknar norr
             ovan, öst höger, söder nedre och väst vänster.\\nSvara på frågan och ange det
             slutliga svaret i formen: '### Svar:'\\n\\nBerättelse:\\n1 XU är diagonalt under XJX
             till höger i 45 graders vinkel.\\n\\nVad är förhållandet från XU till XJX?"}},
4
5        {{"role": "assistant", "content": "### Svar: nedan-höger"}},
6
7        {{"role": "user", "content": "Berättelse:\\n1 XEX är snett ner till höger om XEM.\\n2 XFR
             är ovanför och till höger om XEM.\\n3 XEX är placerad till vänster om XJM.\\n\\nVad är
             förhållandet från XJM till XFR?"}},
8
9        {{"role": "assistant", "content": "### Svar: nedan-höger"}},
10
11
12       {{"role": "user", "content": "Berättelse:\\n1 XAV är till höger om XH.\\n2 XH är till
             höger och XDC är till vänster.\\n3 XAE och XJT är vertikala och XAE är under XJT.\\n4
             XDC är placerat ovanpå XJT.\\n5 XEG är vid 6:00-positionen till XAE.\\n\\nVad är
             förhållandet från XAV till XEG?"}},
13
14       {{"role": "assistant", "content": "### Svar: ovan-höger"}},
15
16
17       {{"role": "user", "content": "Berättelse:\\n1 XBK sitter i höger riktning om XGX.\\n2 XDT
             är ovanför XGX.\\n3 XIC är under och till vänster om XDT.\\n4 XIC och XBD är i en
             vertikal linje med XIC ovanpå.\\n5 XBD är sydost om XFT.\\n6 Om XFT är mitten av en
             urtavla är XDV placerad mellan 7 och 8.\\n7 XDV är placerat längst ner på XFY.\\n\\
             nVad är förhållandet från XFY till XBK?"}},
18
19
20       {{"role": "assistant", "content": "### Svar: vänster"}},
21
22
23       {{"role": "user", "content": "Berättelse:\\n1 XJV och XEJ ligger horisontellt och XJV är
             till vänster av XEJ.\\n2 XJV är direkt nordost om XEX.\\n3 XEX är horisontellt till
             höger om XDU.\\n4 XDU och XCF är vertikala och XDU är ovanför XCF.\\n5 XQ är placerad
             längst upp till vänster om XCF.\\n6 XQ och XGQ är sida vid sida med XQ till höger och
             XGQ till vänster.\\n7 XGQ är där borta med XIY är direkt ovanför.\\n8 Objekten XIY och
              XDV är där borta. Objektet XIY är lägre och något till höger om objektet XDV.\\n9 XCN
              är i 45 graders vinkel mot XDV, i det övre vänstra hörnet.\\n\\nVad är förhållandet
             från XCN till XEJ?"}},
24
25       {{"role": "assistant", "content": "### Svar: vänster"}},
26
27
28       {{"role": "user", "content": "Berättelse:\\n{line['data'].replace('\n', '\\n')}\\n{line['
             question'].replace('\n', '\\n')}"}}]}}
```

## A.6 ASP-Translation Prompt

```
1        {"role": "user", "content": "Given a story about spatial relations among objects, convert
             the relations between objects into facts.\nIf a sentence is describing clock-wise
             information, then 12 denotes top, 1 and 2 denote top_right, 3 denotes right, 4 and 5
             denote down_right, 6 denotes down, 7 and 8 denote down_left, 9 denote left, 10 and 11
             denote top_left. If a sentence is describing cardinal directions, then north denotes
             top, east denotes right, south denotes down, and west denotes left. If a sentence is a
              question, the fact starts with query. Please respect the order of objects in the
             question while generating the query. Otherwise, the fact starts with one of top, down,
              left, right, top_left, top_right, down_left, and down_right.\nJust give me the facts
             and query. No yapping.\n\nStory:\nXAH is positioned in the front right corner of XAM.\
             nXAF is on the left side of and below XAQ.\nXAY and XAI are parallel, and XAY is on
             top of XAI.\nXAV is over there with XAT above.\nXAV is slightly off center to the top
             left and XAG is slightly off center to the bottom right.\nThe objects XAS and XAA are
             over there. The object XAS is lower and slightly to the left of the object XAA.\nXAD
```

is diagonally below XAZ to the right at a 45 degree angle.\nXAV is at XAA\u2019s 9 o\u2019clock\nXAJ is at XAO\u2019s 6 o\u2019clock.\nXAH is below XAJ at 4 o\u2019clock.\nXAO is there and XAC is at the 5 position of a clock face.\nIf XAH is the center of a clock face, XAB is located between 10 and 11.\nWhat is the relation of the agent XAX to the agent XAY?"}, {"role": "assistant", "content": "top_right(\"XAH\", \"XAM\").\ndown_left(\"XAF\", \"XAQ\").\ntop(\"XAY\", \"XAI\").\ntop(\"XAT\", \"XAV\").\ntop_left(\"XAV\", \"XAG\").\ndown_left(\"XAS\", \"XAA\").\ndown_right(\"XAD\", \"XAZ\").\nleft(\"XAV\", \"XAA\").\ndown(\"XAJ\", \"XAO\").\ndown_right(\"XAH\", \"XAJ\").\ndown_right(\"XAC\", \"XAO\").\ntop_left(\"XAB\", \"XAH\").\nquery(\"XAX\", \"XAY\").\n"}, {"role": "user", "content": "Story:\nn1 XIC is placed on the top of XAL.\nWhat is the relation of the agent XIC to the agent XAL?"}]}}

## B  Knowledge Module

```
1    % general format translation, which can also be easily done in python script
2    % (this is not needed if we directly extract the general form in the beginning as in bAbI
         task4)
3    is(A, top, B) :- top(A, B).
4    is(A, top, B) :- up(A, B).
5    is(A, down, B) :- down(A, B).
6    is(A, left, B) :- left(A, B).
7    is(A, right, B) :- right(A, B).
8    is(A, top_left, B) :- top_left(A, B).
9    is(A, top_right, B) :- top_right(A, B).
10   is(A, down_left, B) :- down_left(A, B).
11   is(A, down_right, B) :- down_right(A, B).
12   is(A, east, B) :- east(A, B).
13   is(A, west, B) :- west(A, B).
14   is(A, south, B) :- south(A, B).
15   is(A, north, B) :- north(A, B).
16   % synonyms
17   synonyms(
18       north, northOf; south, southOf; west, westOf ; east, eastOf; top, northOf; down,
            southOf; left, westOf; right, eastOf
19   ).
20   synonyms(A, B) :- synonyms(B, A).
21   synonyms(A, C) :- synonyms(A, B), synonyms(B, C) , A!=C.
22   % define the offsets of 8 spatial relations
23   offset( overlap,0,0; top,0,1; down,0,-1; left,-1,0; right,1,0; top_left,-1,1; top_right
            ,1,1; down_left ,-1,-1; down_right,1,-1 ).
24   % derive the kind of spatial relation from synonyms and offset
25   is(A, R1, B) :- is(A, R2, B), synonyms(R1, R2).
26   is(A, R1, B) :- is(B, R2, A), offset(R2,X,Y), offset(R1,-X,-Y).
27   % derive the location of every object
28   % the search space of X or Y coordinate is within -100 and 100
29   % (to avoid infinite loop in clingo when data has error)
30   nums(-100..100).
31   location(A, Xa, Ya) :- location(B, Xb, Yb), nums(Xa), nums(Ya), is(A, Kind, B), offset(
            Kind, Dx, Dy), Xa-Xb=Dx, Ya-Yb=Dy.
32   location(B, Xb, Yb) :- location(A, Xa, Ya), nums(Xb), nums(Yb), is_on(A, Kind, B), offset(
            Kind, Dx, Dy), Xa-Xb=Dx, Ya-Yb=Dy.
```

## C  Experiment Details

Many state-of-the-art LLMs are now commercially available and we tested a selection: OpenAI GPT 3.5 turbo version 0125, OpenAI GPT-4o version 2024-08-06, OpenAI o1 version 2024-12-17, Open AI o3 mini version 2025-01-31, OpenAI o4-mini version 2025-04-16, Anthropic Claude Sonnet version 20250219, Moonshot AI Kimi-k2, Qwen3-8B, XAI Grok version 2-1212, and Deepseek R1. We used the Microsoft Azure OpenAI service for OpenAI models; other models were provided by their respective vendor's API, except Kimi-k2 and Qwen3 where we used OpenRouter[9]. We switched off the guard rails for models hosted on Azure OpenAI. We set `max_tokens` to 512 in the Anthropic API, it being a required parameter.

---

[9]https://openrouter.ai

Table 4: Experiment-wise Token Usage and Cost Statistics.

| Experiment | Prompt Tokens | Completion Tokens | Input Cost | Output Cost | Total Cost |
|---|---|---|---|---|---|
| 0-Shot Overgeneralisation | 19,889,180 | 28,785,209 | $99.99 | $1,315.90 | $1,415.89 |
| 0-shot Productivity | 5,745,956 | 8,495,821 | $28.88 | $381.43 | $410.31 |
| 0-Shot Substitutivity | 16,904,043 | 9,882,567 | $42.26 | $98.83 | $141.09 |
| 5-shot Overgeneralisation | 48,513,966 | 49,908,106 | $87.18 | $220.51 | $307.69 |
| 5-Shot Productivity | 66,171,935 | 85,971,782 | $195.09 | $1,162.51 | $1,357.59 |
| 5-Shot Substitutivity | 50,467,980 | 26,865,614 | $108.38 | $119.56 | $227.94 |
| Natural Language Generation | 39,062,862 | 25,840,176 | $78.51 | $114.68 | $193.20 |
| Systematicity | 28,601,654 | 26,879,334 | $56.04 | $119.04 | $175.09 |
| ASP Facts | 88,869,452 | 105,909,194 | $54.95 | $145.09 | $200.03 |

Table 5: Total token usage and total cost across all experiments without double-counting reused data points.

| Description | Prompt Tokens | Completion Tokens | Input Cost | Output Cost | Total Cost |
|---|---|---|---|---|---|
| Total Experimental Cost | 327,757,412 | 331,263,618 | $660.04 | $3,172.97 | $3,833.01 |

LLMs are stochastic in nature and show considerable variability in their answers. Vendors provide various API options (e.g. `seed`, `temperature`, and `top_p`) to try to make sampling more consistent. However, no settings that we have yet tried (including setting `temperature` to 0) result in fully deterministic answers. We therefore accept all model defaults and repeat each chat completion multiple times (typically $n = 3$) where we have sufficient data we compute the prediction interval across multiple experimental repeats (Blackwell et al., 2024).

All LLM experiments were conducted using the Golem software[10]. For each prompt we add `Answer the question and provide the final answer in the form:'### Answer:'` to facilitate pattern matching and automation of answer assessment using regular expressions. Each question is presented to the LLM in a separate chat session to avoid cross contamination of answers.

We anonymise the node names using nonce words[11]. The nonce words were generated by randomly sampling a Markov chain model using trigrams of words from Jane Austen's Pride and Prejudice. Nonce words were discarded unless they had seven letters and a Levenshtein edit distance of at least two from all words in the Hunspell English language dictionary[12].

## C.1  Costs

Table 4 presents the experiment-wise cost analysis for all experiments conducted in the main paper. The column *Prompt Token* denotes the total number of input tokens supplied to the LLMs for a given experiment, whereas *Completion Token* represents the total number of output tokens generated collectively by all models within that experiment. These token counts are combined with the corresponding per-million-token pricing for each LLM to compute the *Input Cost* (derived from *Prompt Token*) and the *Output Cost* (derived from *Completion Token*). The *Total Cost* is then calculated as the sum of the *Input Cost* and *Output Cost*. Please note that certain baseline settings (e.g., the 5-shot clean shuffled dataset) are reused across multiple experiments. Consequently, the overall experimental cost cannot be obtained by simply summing the costs of the individual experiments, as this would result in double-counting shared baseline evaluations. Therefore, we computed the total cost across all unique experimental datasets, as reported in Table 5, which amounts to $3,833.01. Among all evaluated models, the highest cost is incurred by the `o1` model, which charges *15.00 per 1M input tokens* and *60.00 per 1M output tokens*.

---

[10]https://github.com/RobBlackwell/golem
[11]"A nonce word (from the 16th-century phrase for the nonce, meaning 'for the once') is a lexeme created for temporary use, to solve an immediate problem of communication." The Cambridge encyclopedia of the English language (Crystal, 2018).
[12]https://github.com/hunspell/hunspell.

Table 6: Productivity experiment results. Comparison of different LLM models for different hops (k) questions based on accuracy (± standard deviation). `CSn3.7`, `DSR1`, `o3m`, `qw8b` represent `C Sonnet` 3.7, DeepSeek R1, `o3 mini`, `qwen3-8b` LLM models respectively.

| $k$ | CSn3.7 | DSR1 | GPT 3.5 | GPT-4o | Grok | kimi-k2 | o1 | o3m | o4-mini | qw8b |
|---|---|---|---|---|---|---|---|---|---|---|
| 1 | $0.98 \pm 0.01$ | 0.99 | $0.73 \pm 0.01$ | $0.99 \pm 0.01$ | 0.97 | $0.97 \pm 0.01$ | $0.98 \pm 0.00$ | 0.98 | $0.99 \pm 0.00$ | 0.81 |
| 2 | $0.77 \pm 0.01$ | 0.67 | $0.20 \pm 0.02$ | $0.60 \pm 0.01$ | 0.51 | $0.50 \pm 0.02$ | $0.65 \pm 0.01$ | 0.76 | $0.83 \pm 0.02$ | 0.40 |
| 3 | $0.53 \pm 0.00$ | 0.67 | $0.16 \pm 0.03$ | $0.35 \pm 0.02$ | 0.38 | $0.31 \pm 0.02$ | $0.58 \pm 0.01$ | 0.61 | $0.75 \pm 0.05$ | 0.42 |
| 4 | $0.41 \pm 0.02$ | 0.65 | $0.12 \pm 0.01$ | $0.29 \pm 0.01$ | 0.29 | $0.29 \pm 0.01$ | $0.35 \pm 0.04$ | 0.62 | $0.70 \pm 0.01$ | 0.41 |
| 5 | $0.37 \pm 0.03$ | 0.62 | $0.12 \pm 0.02$ | $0.29 \pm 0.03$ | 0.31 | $0.25 \pm 0.03$ | $0.21 \pm 0.01$ | 0.51 | $0.65 \pm 0.01$ | 0.34 |
| 6 | $0.26 \pm 0.02$ | 0.61 | $0.15 \pm 0.02$ | $0.27 \pm 0.02$ | 0.27 | $0.24 \pm 0.03$ | $0.12 \pm 0.03$ | 0.53 | $0.62 \pm 0.03$ | 0.31 |
| 7 | $0.22 \pm 0.03$ | 0.54 | $0.14 \pm 0.01$ | $0.20 \pm 0.01$ | 0.21 | $0.20 \pm 0.02$ | $0.04 \pm 0.00$ | 0.49 | $0.59 \pm 0.01$ | 0.23 |
| 8 | $0.22 \pm 0.02$ | 0.48 | $0.12 \pm 0.02$ | $0.22 \pm 0.02$ | 0.26 | $0.18 \pm 0.02$ | $0.04 \pm 0.01$ | 0.43 | $0.58 \pm 0.04$ | 0.23 |
| 9 | $0.20 \pm 0.03$ | 0.37 | $0.11 \pm 0.00$ | $0.19 \pm 0.00$ | 0.18 | $0.18 \pm 0.01$ | $0.02 \pm 0.01$ | 0.39 | $0.52 \pm 0.02$ | 0.21 |
| 10 | $0.20 \pm 0.03$ | 0.34 | $0.12 \pm 0.02$ | $0.21 \pm 0.03$ | 0.17 | $0.17 \pm 0.02$ | $0.04 \pm 0.01$ | 0.43 | $0.52 \pm 0.03$ | 0.16 |
| 20 | $0.12 \pm 0.01$ | 0.14 | $0.15 \pm 0.02$ | $0.15 \pm 0.01$ | 0.13 | $0.14 \pm 0.01$ | $0.00 \pm 0.01$ | 0.17 | $0.36 \pm 0.04$ | 0.10 |
| 50 | $0.10 \pm 0.01$ | 0.17 | $0.12 \pm 0.02$ | $0.18 \pm 0.01$ | 0.20 | $0.12 \pm 0.00$ | $0.01 \pm 0.00$ | 0.15 | $0.17 \pm 0.01$ | 0.09 |
| 100 | $0.09 \pm 0.01$ | 0.14 | $0.12 \pm 0.02$ | $0.15 \pm 0.02$ | 0.17 | $0.13 \pm 0.01$ | $0.01 \pm 0.00$ | 0.08 | $0.07 \pm 0.00$ | 0.10 |

## D  Productivity Experiment

Results in Table 6 are supplementary to results already presented in Figure 2. In addition to the results presented for all the LLM models for specific $k = 1, 2, 10$ in the Figure 2, we present here the results for all values of $k$. As we can observe in Table 6, there is a sudden drop in performance from $k = 1$ to $k = 2$ which can be attributed to the fact that the reasoning of the models begins from $k = 2$. We further notice that all the LLMs (`GPT 3.5`, `GPT-4o`, `Grok`, `C Sonnet 3.7`, `o4-mini`) exhibit a sharp performance collapse at lower values of k, whereas all the LRM models (DeepSeek `R1`, `o3 mini`, `qwen3-8b`, `o1` ) show a more gradual decline in performance, which we attribute to their stronger built-in reasoning capabilities. Please note that we take a strict definition that a LRM is a model that outputs a reasoning trace or reports number of reasoning tokens used $> 0$. Also, if a model has typically been defined as LRM, but we ran it in the standard mode (without reasoning), we will categorize the model as LLM. We conducted further analysis of the types of errors made by the models. For $k = 1$, we observed that most models tended to produce incorrect answers on the same set of questions, indicating a high degree of overlap in failure cases. However, as the value of $k$ increased, this pattern of shared errors diminished, and no consistent similarity was observed in the error distributions across either the LLM or LRM models.

We perform further analysis in Table 8 that summarizes the accuracy degradation from $k = 1$ to $k = 2$ for the models evaluated in Table 6. The *Accuracy Drop* column reports the transition in accuracy in the form $k = 1 \to k = 2$ followed by the absolute decrease in accuracy shown in parentheses. The largest degradation is observed for either older-generation models (e.g., `GPT 3.5`) or models that are not explicitly designed as reasoning-first models (e.g., `kimi-k2` and `Grok`). In contrast, models explicitly optimized for reasoning exhibit comparatively smaller declines in performance, including `o1`, `DSR1` (DeepSeek-R1), `o3m` (o3-mini), `CSn3.7` (Claude Sonnet 3.7), and `o4-mini`. We further manually inspected the predictions at $k = 2$ across all evaluated models and did not observe any additional anomalous behaviour.

We further conducted experiments to compare the performance of LLMs under zero-shot and five-shot In-Context Learning (ICL) settings. The results, presented in Table 7, include evaluations for `GPT-4o`, `o1`, and `llama-3-70b-instruct` across values of $k = 1, 2, \ldots, 10, 20, 50, 100$. For the five-shot ICL setting, we use prompts containing *shuffled* stories with $k = 1, 3, 5, 7, 10$. Overall, models such as `GPT-4o` and `llama-3-70b-instruct` consistently demonstrate improved performance when provided with in-context examples in the prompt compared to the zero-shot setting. However,the behavior of the `o1` model is counter-intuitive. Our further analysis of the results indicate that `o1` allows *other* label that typically corresponds to responses such as *not enough information to determine*, *cannot be determined from the given information* or *the information in the story is insufficient to determine XHX's exact relation to XN*. The label *other* is predicted significantly

Table 7: Productivity experiment comparing the results for zero In-Context Learning examples and 5 In-Context Learning examples. The table shows the comparison of different LLM models for different hops (k) questions based on accuracy ($\pm$ standard deviation). `llama-3-70b-i` represents `llama-3-70b-instruct` model, `0-ICL` and `5-ICL` represent zero In-Context Learning and five In-Context Learning examples provided to the model.

| $k$ | GPT-4o | | o1 | | llama-3-70b-i | |
|---|---|---|---|---|---|---|
| | 0-ICL | 5-ICL | 0-ICL | 5-ICL | 0-ICL | 5-ICL |
| 1 | $0.98 \pm 0.01$ | $0.99 \pm 0.01$ | 0.98 | 0.98 | 0.88 | 0.97 |
| 2 | $0.59 \pm 0.02$ | $0.60 \pm 0.01$ | 0.71 | 0.67 | 0.36 | 0.44 |
| 3 | $0.32 \pm 0.03$ | $0.35 \pm 0.02$ | 0.67 | 0.58 | 0.27 | 0.27 |
| 4 | $0.27 \pm 0.01$ | $0.29 \pm 0.01$ | 0.54 | 0.33 | 0.24 | 0.23 |
| 5 | $0.28 \pm 0.01$ | $0.29 \pm 0.03$ | 0.47 | 0.21 | 0.24 | 0.31 |
| 6 | $0.22 \pm 0.01$ | $0.27 \pm 0.02$ | 0.40 | 0.10 | 0.19 | 0.24 |
| 7 | $0.19 \pm 0.02$ | $0.20 \pm 0.01$ | 0.20 | 0.04 | 0.17 | 0.20 |
| 8 | $0.19 \pm 0.01$ | $0.22 \pm 0.02$ | 0.15 | 0.03 | 0.14 | 0.18 |
| 9 | $0.19 \pm 0.03$ | $0.19 \pm 0.00$ | 0.08 | 0.03 | 0.17 | 0.20 |
| 10 | $0.17 \pm 0.02$ | $0.21 \pm 0.03$ | 0.10 | 0.04 | 0.14 | 0.18 |
| 20 | $0.12 \pm 0.02$ | $0.15 \pm 0.01$ | 0.01 | 0.01 | 0.13 | 0.14 |
| 50 | $0.15 \pm 0.02$ | $0.18 \pm 0.01$ | 0.00 | 0.01 | 0.15 | 0.15 |
| 100 | $0.07 \pm 0.02$ | $0.15 \pm 0.02$ | 0.00 | 0.01 | 0.13 | 0.14 |

| Model | Accuracy Drop | Model Size | Model Type | Reasoning | Release Date |
|---|---|---|---|---|---|
| GPT 3.5 | $0.73 \rightarrow 0.20\ (0.53)$ | – | Closed-source | No | January, 2024 |
| kimi-k2 | $0.97 \rightarrow 0.50\ (0.47)$ | 1T | Open-weight | No | July, 2025 |
| Grok | $0.97 \rightarrow 0.51\ (0.46)$ | 270B | Open-weight | No | December 2024 |
| qw8b | $0.81 \rightarrow 0.40\ (0.41)$ | 8B | Open-weight | Yes | April, 2025 |
| GPT-4o | $0.99 \rightarrow 0.60\ (0.39)$ | – | Closed-source | No | August, 2024 |
| o1 | $0.98 \rightarrow 0.65\ (0.33)$ | – | Closed-source | Yes | December, 2024 |
| DSR1 | $0.99 \rightarrow 0.67\ (0.32)$ | 671B | Open-weight | Yes | January, 2025 |
| o3m | $0.98 \rightarrow 0.76\ (0.22)$ | – | Closed-source | Yes | January, 2025 |
| CSn3.7 | $0.98 \rightarrow 0.77\ (0.21)$ | – | Closed-source | Yes | February, 2025 |
| o4 mini | $0.99 \rightarrow 0.83\ (0.16)$ | – | Closed-source | Yes | April, 2025 |

Table 8: Comparison of model performance degradation (derived from Table 6) based on model characteristics, and release dates. The value in parentheses denotes the absolute accuracy drop. `CSn3.7, DSR1, o3m, qw8b` represent Claude Sonnet 3.7, DeepSeek R1, o3 mini, qwen3 8B LLM models respectively.

more frequently in the 5-shot setting compared to the 0-shot setting in `o1`. This suggests that providing five in-context examples may *prime* the model to express uncertainty more explicitly when it is unsure about the correct answer. In contrast, the 0-shot setting appears to *encourage* the model to commit to a specific answer, even in the absence of sufficient information, thereby reducing the frequency of such responses.

# E  Systematicity Experiment

Results in Table 9 are supplementary to the results already presented in Table 1 in the main paper. Here we present results for $k = 3, 4, 6, 7, 8, 9, 20, 50, 100$ as additional results for analysing systematicity by replacing English language by Nonsense language. These experiments are performed thrice, using the `GPT-4o` model to ensure reproducibility of the results. We also have results for `o4-mini` where the experiments for English are performed thrice whereas the experiments for Nonce language are performed only once. As can be seen from the table, the experiment exhibits the same pattern for $k = 20, 50, 100$ as it does for $k = 1, 2, 5, 10$: the performance of model on nonce-instance results remains lower than English results even at the higher values of $k$. This shows that LLM lack systematicity to reason in this spatial domain.

| $k$ | GPT-4o | | o4-mini | |
|---|---|---|---|---|
| | English | Nonce | English | Nonce |
| 1 | $0.99 \pm 0.01$ | $0.37 \pm 0.01$ | $0.99 \pm 0.00$ | 0.51 |
| 2 | $0.60 \pm 0.01$ | $0.14 \pm 0.01$ | $0.83 \pm 0.02$ | 0.38 |
| 3 | $0.35 \pm 0.02$ | $0.14 \pm 0.03$ | $0.75 \pm 0.05$ | 0.42 |
| 4 | $0.29 \pm 0.01$ | $0.16 \pm 0.03$ | $0.70 \pm 0.01$ | 0.34 |
| 5 | $0.29 \pm 0.03$ | $0.14 \pm 0.01$ | $0.65 \pm 0.01$ | 0.34 |
| 6 | $0.27 \pm 0.02$ | $0.15 \pm 0.02$ | $0.62 \pm 0.03$ | 0.31 |
| 7 | $0.20 \pm 0.01$ | $0.13 \pm 0.03$ | $0.59 \pm 0.01$ | 0.29 |
| 8 | $0.22 \pm 0.02$ | $0.17 \pm 0.03$ | $0.58 \pm 0.04$ | 0.28 |
| 9 | $0.19 \pm 0.00$ | $0.14 \pm 0.02$ | $0.52 \pm 0.02$ | 0.28 |
| 10 | $0.21 \pm 0.03$ | $0.10 \pm 0.02$ | $0.52 \pm 0.03$ | 0.26 |
| 20 | $0.15 \pm 0.01$ | $0.15 \pm 0.04$ | $0.36 \pm 0.04$ | 0.20 |
| 50 | $0.18 \pm 0.01$ | $0.14 \pm 0.05$ | $0.17 \pm 0.01$ | 0.10 |
| 100 | $0.15 \pm 0.02$ | $0.11 \pm 0.01$ | $0.07 \pm 0.00$ | 0.14 |

Table 9: Mean accuracy ($\pm$ standard deviation) for English and nonsense stories across GPT-4o and o4-mini under 5-shot ICL prompting.

# F   Substitutivity: Natural Language Translation Experiment

| $k$ | GPT-4o | | | o4-mini | | |
|---|---|---|---|---|---|---|
| | English | Hindi | Swedish | English | Hindi | Swedish |
| 1 | $0.99 \pm 0.01$ | $0.96 \pm 0.01$ | $0.82 \pm 0.01$ | $0.99 \pm 0.00$ | 0.95 | 0.30 |
| 2 | $0.60 \pm 0.01$ | $0.47 \pm 0.02$ | $0.43 \pm 0.01$ | $0.83 \pm 0.02$ | 0.73 | 0.23 |
| 3 | $0.35 \pm 0.02$ | $0.29 \pm 0.02$ | $0.34 \pm 0.02$ | $0.75 \pm 0.05$ | 0.68 | 0.14 |
| 4 | $0.29 \pm 0.01$ | $0.20 \pm 0.02$ | $0.29 \pm 0.02$ | $0.70 \pm 0.01$ | 0.62 | 0.11 |
| 5 | $0.29 \pm 0.03$ | $0.22 \pm 0.01$ | $0.26 \pm 0.01$ | $0.65 \pm 0.01$ | 0.63 | 0.09 |
| 6 | $0.27 \pm 0.02$ | $0.22 \pm 0.03$ | $0.23 \pm 0.01$ | $0.62 \pm 0.03$ | 0.54 | 0.07 |
| 7 | $0.20 \pm 0.01$ | $0.15 \pm 0.01$ | $0.21 \pm 0.00$ | $0.59 \pm 0.01$ | 0.62 | 0.05 |
| 8 | $0.22 \pm 0.02$ | $0.17 \pm 0.03$ | $0.21 \pm 0.02$ | $0.58 \pm 0.04$ | 0.58 | 0.10 |
| 9 | $0.19 \pm 0.00$ | $0.18 \pm 0.01$ | $0.18 \pm 0.01$ | $0.52 \pm 0.02$ | 0.49 | 0.08 |
| 10 | $0.21 \pm 0.03$ | $0.18 \pm 0.01$ | $0.20 \pm 0.00$ | $0.52 \pm 0.03$ | 0.48 | 0.07 |
| 20 | $0.15 \pm 0.01$ | $0.15 \pm 0.01$ | $0.15 \pm 0.01$ | $0.36 \pm 0.04$ | 0.28 | 0.05 |
| 50 | $0.18 \pm 0.01$ | $0.13 \pm 0.04$ | $0.15 \pm 0.01$ | $0.17 \pm 0.01$ | 0.14 | 0.15 |
| 100 | $0.15 \pm 0.02$ | $0.13 \pm 0.01$ | $0.15 \pm 0.02$ | $0.07 \pm 0.00$ | 0.06 | 0.07 |

Table 10: Mean accuracy ($\pm$ standard deviation) across languages (English, Hindi, Swedish) for GPT-4o and o4-mini under 5-shot ICL.

Results in Table 10 are supplementary to the results already presented in Table 2 in the main paper. Here we present results for $k = 3, 4, 6, 7, 8, 9, 20, 50, 100$ as additional results for analysing the natural language translation experiments. The results are obtained using the `GPT-4o` model by performing 3 repeats to ensure reproducibility. We report mean accuracy and standard deviation across three runs.

We also present the results for `o4-mini` where results for English are repeated three times but the results for Hindi and Swedish are obtained only once due to budget constraints. Because `o4-mini` is a more recent model than `GPT-4o` and has been specifically optimised for reasoning[13], its overall performance is better than that of `GPT-4o`. Further, the performance of `o4-mini` degrades more gradually with increasing values of $k$ compared to `GPT-4o` due to its stronger reasoning capabilities for longer multi-hop reasoning chains.

---

[13]https://openai.com/index/introducing-o3-and-o4-mini/ (last accessed 9 May 2025)

# G    Symbolic Evaluation of LLMs

Results in Table 11 are supplementary to the results already presented in Table 3 in the main paper. Here we present results for remaining values of $k$ as additional results for analysing the translation of natural language sentences into ASP facts. As discussed earlier, the prompt used to generate the answers is shown in detail in Appendix A.6. The results are obtained by running one repeat of the experiment with `GPT-4o` model because of the resources limitations. The results under `LLM+ASP` indicate that the translation of text sentences to ASP facts starts to deteriorate as $k$ increases highlighting LLMs inability to translate the natural language text to ASP facts. As already discussed in Section 3 of the paper, `Oracle+ASP` translates natural language sentences from the dataset into ASP facts (gold label) and attach predefined ASP knowledge module (defined in Section B) and the resulting ASP program is evaluated using Clingo to derive the answer. This baseline serves as a verification step to ensure the correctness of the generated data.

| | | $k \rightarrow$ | | | | | | | | | | | | |
|---|---|---|---|---|---|---|---|---|---|---|---|---|---|---|
| Type | Model | 1 | 2 | 3 | 4 | 5 | 6 | 7 | 8 | 9 | 10 | 20 | 50 | 100 |
| LLM + ASP | GPT-4o | 0.9 | 0.9 | 1.0 | 0.9 | 0.9 | 0.9 | 0.9 | 0.9 | 0.9 | 0.9 | 0.8 | 0.6 | 0.2 |
| | Gemini 2.5 Flash | 1.0 | 1.0 | 1.0 | 1.0 | 1.0 | 1.0 | 1.0 | 0.9 | 1.0 | 0.9 | 0.9 | 0.7 | 0.5 |
| | o4-mini | 1.0 | 1.0 | 1.0 | 0.9 | 0.9 | 0.9 | 0.9 | 0.9 | 0.9 | 0.9 | 0.8 | 0.7 | 0.5 |
| | gpt5.1 | 0.9 | 0.8 | 0.9 | 0.9 | 0.9 | 0.9 | 0.9 | 0.9 | 0.8 | 0.8 | 0.7 | 0.5 | 0.3 |
| | Kimi K2 | 0.4 | 0.3 | 0.2 | 0.1 | 0.2 | 0.2 | 0.2 | 0.2 | 0.2 | 0.2 | 0.3 | 0.1 | 0.1 |
| | DeepSeek | 0.8 | 0.9 | 0.9 | 0.9 | 0.9 | 0.9 | 0.9 | 0.9 | 0.8 | 0.8 | 0.7 | 0.6 | 0.4 |
| | GPT OSS | 0.8 | 0.8 | 0.8 | 0.8 | 0.8 | 0.8 | 0.8 | 0.8 | 0.7 | 0.8 | 0.7 | 0.5 | 0.3 |
| Oracle + ASP | | 1.0 | 1.0 | 1.0 | 1.0 | 1.0 | 1.0 | 1.0 | 1.0 | 1.0 | 1.0 | 1.0 | 1.0 | 1.0 |

Table 11: Supplementary results for the ASP translation experiment in Section 4.5, showing accuracy by $k$ for ASP runs over all samples.

# H    Overgeneralisation Experiment

Table 12 presents supplementary overgeneralization results that complement the findings shown in Figure 3 of the main paper. In addition to the 5-shot in-context learning (ICL) overgeneralization results reported for `GPT-4o`, we also include corresponding results for `o4-mini`. Consistent with the analysis in the main paper, we compare model performance under *shuffled* versus *ordered* stories (both clean), as well as *clean* versus *noisy* story (both ordered) settings.

The comparison between `GPT-4o` and `o4-mini` indicates that `o4-mini` consistently achieves stronger performance across most settings. Moreover, unlike `GPT-4o`, which exhibits a sharp decline in performance as the number of hops ($k$) increases, `o4-mini` demonstrates a more gradual degradation, suggesting greater robustness to increasing reasoning complexity. Additionally, the results show that ordered stories generally yield better performance than shuffled stories, while clean stories outperform noisy stories across both models.

Table 13 presents the overgeneralisation results under a 0-shot in-context learning (ICL) setting for the `GPT-4o`, `Llama-3-70B`, and `o1` models. Similar to 5-shot ICL experiment, for each model, we report comparative accuracy across two dimensions: input ordering (ordered vs. shuffled) and data quality (clean vs. noisy). These results serve as supplementary evidence, complementing the trends in Figure 4 in the main paper.

Overall, a consistent pattern is observed across all models: performance is higher when the input stories are ordered rather than shuffled, and when the inputs are clean rather than noisy. This behaviour aligns with expectations, as both story order and presence of noisy sentences are known to influence reasoning performance as established in the main paper.

In terms of relative performance, the `o1` model achieves the highest accuracy, followed by `GPT-4o`, while `Llama-3-70B` exhibits comparatively lower performance across settings.

| $k$ | GPT-4o | | o4-mini | | GPT-4o | | o4-mini | |
|---|---|---|---|---|---|---|---|---|
| | Ordered | Shuffled | Ordered | Shuffled | Clean | Noisy | Clean | Noisy |
| 1 | 0.99 | 0.98 | 0.99 | 0.99 | 0.99 | 0.92 | 0.99 | 0.99 |
| 2 | 0.57 | 0.55 | 0.89 | 0.86 | 0.57 | 0.47 | 0.89 | 0.87 |
| 3 | 0.34 | 0.40 | 0.83 | 0.82 | 0.34 | 0.31 | 0.83 | 0.84 |
| 4 | 0.34 | 0.27 | 0.82 | 0.79 | 0.34 | 0.29 | 0.82 | 0.77 |
| 5 | 0.39 | 0.29 | 0.83 | 0.74 | 0.39 | 0.30 | 0.83 | 0.76 |
| 6 | 0.31 | 0.23 | 0.80 | 0.71 | 0.31 | 0.29 | 0.80 | 0.77 |
| 7 | 0.28 | 0.21 | 0.84 | 0.69 | 0.28 | 0.26 | 0.84 | 0.78 |
| 8 | 0.27 | 0.23 | 0.79 | 0.68 | 0.27 | 0.23 | 0.79 | 0.74 |
| 9 | 0.26 | 0.20 | 0.81 | 0.63 | 0.26 | 0.22 | 0.81 | 0.70 |
| 10 | 0.24 | 0.20 | 0.80 | 0.65 | 0.24 | 0.22 | 0.80 | 0.73 |
| 20 | 0.16 | 0.14 | 0.77 | 0.48 | 0.16 | 0.15 | 0.77 | 0.61 |
| 50 | 0.18 | 0.19 | 0.69 | 0.16 | 0.18 | 0.14 | 0.69 | 0.39 |
| 100 | 0.14 | 0.16 | 0.39 | 0.07 | 0.14 | 0.17 | 0.39 | 0.20 |

Table 12: Overgeneralisation accuracy under ordered vs. shuffled data and clean vs. noisy data (5-shot ICL) for GPT-4o and o4-mini.

| $k$ | GPT-4o | | Llama-3-70B | | o1 | | GPT-4o | | Llama-3-70B | | o1 | |
|---|---|---|---|---|---|---|---|---|---|---|---|---|
| | Ord | Shuff | Ord | Shuff | Ord | Shuff | Clean | Noisy | Clean | Noisy | Clean | Noisy |
| 1 | 0.98 | 0.98 | 0.86 | 0.88 | 0.98 | 0.98 | 0.98 | 0.98 | 0.86 | 0.81 | 0.98 | 0.99 |
| 2 | 0.59 | 0.59 | 0.39 | 0.36 | 0.73 | 0.71 | 0.59 | 0.53 | 0.39 | 0.32 | 0.73 | 0.68 |
| 3 | 0.34 | 0.32 | 0.24 | 0.27 | 0.66 | 0.67 | 0.34 | 0.33 | 0.24 | 0.28 | 0.66 | 0.60 |
| 4 | 0.30 | 0.27 | 0.23 | 0.24 | 0.65 | 0.54 | 0.30 | 0.29 | 0.23 | 0.21 | 0.65 | 0.51 |
| 5 | 0.32 | 0.28 | 0.27 | 0.24 | 0.60 | 0.47 | 0.32 | 0.26 | 0.27 | 0.24 | 0.60 | 0.36 |
| 6 | 0.26 | 0.22 | 0.21 | 0.19 | 0.54 | 0.40 | 0.26 | 0.22 | 0.21 | 0.22 | 0.54 | 0.29 |
| 7 | 0.21 | 0.19 | 0.17 | 0.17 | 0.48 | 0.20 | 0.21 | 0.19 | 0.17 | 0.16 | 0.48 | 0.23 |
| 8 | 0.24 | 0.19 | 0.18 | 0.14 | 0.46 | 0.15 | 0.24 | 0.18 | 0.18 | 0.17 | 0.46 | 0.17 |
| 9 | 0.20 | 0.19 | 0.20 | 0.17 | 0.42 | 0.08 | 0.20 | 0.18 | 0.20 | 0.14 | 0.42 | 0.15 |
| 10 | 0.20 | 0.17 | 0.17 | 0.14 | 0.39 | 0.10 | 0.20 | 0.15 | 0.17 | 0.18 | 0.39 | 0.17 |
| 20 | 0.11 | 0.12 | 0.14 | 0.13 | 0.18 | 0.01 | 0.11 | 0.12 | 0.14 | 0.11 | 0.18 | 0.04 |
| 50 | 0.14 | 0.15 | 0.17 | 0.15 | 0.03 | 0.00 | 0.14 | 0.12 | 0.17 | 0.14 | 0.03 | 0.01 |
| 100 | 0.09 | 0.07 | 0.14 | 0.13 | 0.01 | 0.00 | 0.09 | 0.10 | 0.14 | 0.10 | 0.01 | 0.00 |

Table 13: Overgeneralisation accuracy under ordered (Ord) vs. shuffled (Shuff) data and clean vs. noisy data (0-shot ICL) for GPT-4o, Llama-3-70B and o1 models.

# I  Token Analysis for Natural Language Translation

Table 14 presents the token statistics for the natural language translation experiments across English, Hindi, and Swedish for the `GPT-4o` and `o4-mini` models. `apt` denotes the average number of input prompt tokens per value of $k$, `act` denotes the average number of output tokens generated by the model as the answer, and `att` represents the total average tokens, computed as the sum of `apt` and `act`. As observed from the table, the Hindi natural language translation setting consumes the highest number of input prompt tokens, followed by Swedish and then English. A similar trend is observed for the output tokens, where Hindi requires the largest number of tokens to generate the answer facts, followed by Swedish and English. This behaviour remains consistent across all values of $k$. These findings demonstrate that the computational characteristics of the natural language translation experiment are significantly influenced by the choice of language used in the analysis.

| Model | $k$ | English | | | Hindi | | | Swedish | | |
|---|---|---|---|---|---|---|---|---|---|---|
| | | apt | act | att | apt | act | att | apt | act | att |
| GPT-4o | 1 | 732.06 | 5.50 | 737.56 | 914.71 | 7.12 | 921.83 | 798.75 | 7.89 | 806.63 |
| | 2 | 746.98 | 5.81 | 752.79 | 932.22 | 7.61 | 939.83 | 814.82 | 8.60 | 823.42 |
| | 3 | 763.29 | 5.83 | 769.13 | 951.11 | 7.55 | 958.66 | 832.11 | 8.71 | 840.83 |
| | 4 | 777.42 | 5.80 | 783.23 | 967.77 | 7.67 | 975.43 | 847.54 | 8.77 | 856.32 |
| | 5 | 793.67 | 5.83 | 799.50 | 985.97 | 7.60 | 993.57 | 864.34 | 8.63 | 872.97 |
| | 6 | 807.17 | 5.84 | 813.01 | 1002.33 | 7.69 | 1010.02 | 882.53 | 8.73 | 891.26 |
| | 7 | 823.67 | 5.78 | 829.46 | 1021.18 | 7.56 | 1028.74 | 899.08 | 8.65 | 907.73 |
| | 8 | 837.09 | 5.79 | 842.88 | 1038.26 | 7.64 | 1045.89 | 916.09 | 8.64 | 924.73 |
| | 9 | 852.62 | 5.78 | 858.39 | 1055.61 | 7.74 | 1063.35 | 931.63 | 8.57 | 940.21 |
| | 10 | 869.41 | 5.76 | 875.18 | 1075.70 | 7.42 | 1083.12 | 947.33 | 8.62 | 955.95 |
| | 20 | 1022.35 | 5.76 | 1028.11 | 1253.69 | 7.66 | 1261.36 | 1114.46 | 9.48 | 1123.95 |
| | 50 | 1481.54 | 8.77 | 1490.31 | 1790.52 | 7.38 | 1797.90 | 1608.62 | 8.47 | 1617.10 |
| | 100 | 2235.74 | 14.80 | 2250.55 | 2673.74 | 6.88 | 2680.62 | 2447.70 | 9.51 | 2457.21 |
| o4-mini | 1 | 741.06 | 122.06 | 863.12 | 923.71 | 235.19 | 1158.89 | 807.75 | 305.82 | 1113.57 |
| | 2 | 755.98 | 593.89 | 1349.87 | 941.22 | 756.08 | 1697.30 | 823.82 | 703.14 | 1526.96 |
| | 3 | 772.29 | 901.12 | 1673.41 | 960.11 | 1229.36 | 2189.47 | 841.11 | 1246.19 | 2087.30 |
| | 4 | 786.42 | 1146.66 | 1933.09 | 976.77 | 1480.52 | 2457.29 | 856.54 | 1600.59 | 2457.14 |
| | 5 | 802.67 | 1444.58 | 2247.25 | 994.97 | 1710.12 | 2705.09 | 873.34 | 1673.56 | 2546.89 |
| | 6 | 816.17 | 1548.71 | 2364.89 | 1011.33 | 1844.13 | 2855.47 | 891.53 | 2026.19 | 2917.72 |
| | 7 | 832.67 | 1765.20 | 2597.88 | 1030.18 | 2015.41 | 3045.59 | 908.08 | 2164.84 | 3072.92 |
| | 8 | 846.09 | 1953.76 | 2799.84 | 1047.26 | 2207.59 | 3254.84 | 925.09 | 2522.64 | 3447.72 |
| | 9 | 861.62 | 2254.37 | 3115.99 | 1064.61 | 2387.74 | 3452.35 | 940.63 | 2606.43 | 3547.06 |
| | 10 | 878.41 | 2276.84 | 3155.26 | 1084.70 | 2534.65 | 3619.36 | 956.33 | 2858.24 | 3814.57 |
| | 20 | 1031.35 | 3250.14 | 4281.48 | 1262.69 | 3907.49 | 5170.18 | 1123.46 | 4367.20 | 5490.67 |
| | 50 | 1490.54 | 3418.09 | 4908.62 | 1799.52 | 3400.28 | 5199.80 | 1617.62 | 4312.95 | 5930.57 |
| | 100 | 2244.74 | 3068.43 | 5313.18 | 2682.74 | 2970.30 | 5653.03 | 2456.70 | 3556.64 | 6013.34 |

Table 14: Average token statistics across English, Hindi, and Swedish languages for Natural Language Translation. apt, act represents avg prompt tokens, avg completion tokens, avg total tokens respectively.

## J    Qualitative Analysis of Correct and Incorrect Predictions

In this section, we present a qualitative analysis of the predictions generated by three LLMs: DeepSeek `R1` (Table 15), `GPT-4o` (Table 16), and `o4-mini` (Table 17) for the productivity experiment performed in Section 4.1. For each model, we consider reasoning chains with the number of hops ($k$) ranging from 1 to 5. For every value of $k$, we randomly select one correctly classified instance and one incorrectly classified instance produced by the corresponding LLM.

As described in Section 3, each instance is represented as a triple $\langle s, q, a \rangle$, where $s$ denotes the **story**, $q$ denotes the **question**, and $a$ denotes the **answer**. In the tables, the keyword `Story` represents the combined $(s, q)$ pair, while the true answer $a$ is denoted by `Ground Truth`. The prediction generated by the LLM is represented using the keyword `Predicted Label`. The possible ground-truth labels include *above*, *below*, *left*, *right*, *lower-left*, *lower-right*, *upper-left*, and *upper-right*.

A notable observation from the incorrectly classified examples is that, in many cases, the models still predict a partially correct spatial relation. For example, instead of predicting *upper-right*, the model predicts *right*, or instead of *lower-right*, it predicts *upper-right*. This suggests that the models are often able to capture coarse-grained directional relationships while failing to infer the complete compositional spatial relation accurately.

Table 15: Correctly and Incorrectly Classified Instances by Deepseek R1

| $k$ | Correctly Classified Instances | Incorrectly Classified Instances |
|---|---|---|
| 1 | **Story**:
1 XJI is to the left of XAN and below XAN at approximately a 45 degree angle.
What is the relation of the agent XAN to the agent XJI?

**Predicted Label**: upper-right
**Ground Truth**: upper-right | **Story**:
1 If XBK is the center of a clock face, XHZ is located between 2 and 3.
What is the relation of the agent XBK to the agent XHZ?

**Predicted Label**: upper-right
**Ground Truth**: lower-left |
| 2 | **Story**:
1 XJT is to the right of XFP horizontally.
2 XFP is there and XBC is at the 10 position of a clock face.
What is the relation of the agent XBC to the agent XJT?

**Predicted Label**: upper-left
**Ground Truth**: upper-left | **Story**:
1 XIG is at XJE's 6 o'clock.
2 The object XEL is upper and slightly to the right of the object XIG.
What is the relation of the agent XEL to the agent XJE?

**Predicted Label**: upper-right
**Ground Truth**: right |
| 3 | **Story**:
1 XFM is over there with XFG above.
2 XFM is diagonally right and above XCA.
3 The object labeled XFG is positioned to the left of the object labeled XFA.
What is the relation of the agent XCA to the agent XFA?

**Predicted Label**: lower-left
**Ground Truth**: lower-left | **Story**:
1 XAM is to the right of XDB horizontally.
2 XFO is below XDQ at 7 o'clock.
3 XDQ is on the same vertical plane directly above XDB.
What is the relation of the agent XAM to the agent XFO?

**Predicted Label**: lower-right
**Ground Truth**: right |
| 4 | **Story**:
1 XAD is diagonally to the bottom left of XFD.
2 XAD and XFB are in a vertical line with XAD on top.
3 XCX is positioned in the lower right corner of XFB.
4 XCX is sitting at the lower position to XIC.
What is the relation of the agent XIC to the agent XFD?

**Predicted Label**: below
**Ground Truth**: below | **Story**:
1 XAJ and XEK are side by side with XAJ to the right and XEK to the left.
2 XGG and XEK are parallel, and XEK on the right of XGG.
3 XBC is to the left and above XGG at an angle of about 45 degrees.
4 XFC is below XBC with a small gap between them.
What is the relation of the agent XAJ to the agent XFC?

**Predicted Label**: lower-left
**Ground Truth**: right |

| $k$ | Correctly Classified Instances | Incorrectly Classified Instances |
|---|---|---|
| 5 | **Story**: 
 1 XAZ is sitting at the top position to XCF. 
 2 XT is diagonally to the upper right of XGM. 
 3 XT and XCW are horizontal and XCW is to the right of XT. 
 4 XGM is on the same vertical plane directly below XCM. 
 5 The objects XAZ and XCW are over there. The object XAZ is above and slightly to the right of the object XCW. 
 What is the relation of the agent XCF to the agent XCM? 

 **Predicted Label**: right 
 **Ground Truth**: right | **Story**: 
 1 XAZ is north east of XIO. 
 2 XIO is on the right side to XBS. 
 3 XHL is positioned below XAZ and to the right. 
 4 XDR and XHL are parallel, and XDR is on top of XHL. 
 5 The objects XBS and XEG are over there. The object XBS is lower and slightly to the left of the object XEG. 
 What is the relation of the agent XDR to the agent XEG? 

 **Predicted Label**: lower-right 
 **Ground Truth**: right |

Table 16: Correctly and Incorrectly Classified Instances by GPT-4o

| $k$ | Correctly Classified Instances | Incorrectly Classified Instances |
|---|---|---|
| 1 | **Story**: 
 1 XIZ and XJE are in a horizontal line with XJE on the right. 
 What is the relation of the agent XIZ to the agent XJE? 

 **Predicted Label**: left 
 **Ground Truth**: left | **Story**: 
 1 XGD is positioned in the front right corner of XA. 
 What is the relation of the agent XGD to the agent XA? 

 **Predicted Label**: lower-right 
 **Ground Truth**: upper-right |
| 2 | **Story**: 
 1 The object labeled XEG is positioned to the left of the object labeled XJK. 
 2 XA is diagonally above XJK to the right at a 45 degree. 
 What is the relation of the agent XA to the agent XEG? 

 **Predicted Label**: upper-right 
 **Ground Truth**: upper-right | **Story**: 
 1 XAW is positioned in the top left corner of XDU. 
 2 XAF is on the right and XAW is on the left. 
 What is the relation of the agent XDU to the agent XAF? 

 **Predicted Label**: left 
 **Ground Truth**: below |

| $k$ | Correctly Classified Instances | Incorrectly Classified Instances |
|---|---|---|
| 3 | **Story**:
1 XZ is placed at the upper right of XHW.
2 XHW is at a 45 degree angle to XHF, in the upper lefthand corner.
3 XBV is on the lower left of XHF.
What is the relation of the agent XBV to the agent XZ?

**Predicted Label**: lower-left
**Ground Truth**: lower-left | **Story**:
1 XDB is to the right of XBM with a small gap between them.
2 XJT is there and XDB is at the 8 position of a clock face.
3 XJT is directly north west of XCA.
What is the relation of the agent XBM to the agent XCA?

**Predicted Label**: upper-left
**Ground Truth**: left |
| 4 | **Story**:
1 XDX is to the bottom-left of XBL.
2 XBL is to the left of XN horizontally.
3 XEU is at the lower side of XJF.
4 XEU is below XDX and to the left of XDX.
What is the relation of the agent XN to the agent XJF?

**Predicted Label**: upper-right
**Ground Truth**: upper-right | **Story**:
1 XCD is to the top-right of XS.
2 XJO is above XGC at 10 o'clock.
3 The object XCD is positioned directly above the object XGI.
4 XS is diagonally to the upper right of XGC.
What is the relation of the agent XGI to the agent XJO?

**Predicted Label**: below
**Ground Truth**: right |
| 5 | **Story**:
1 XBZ is positioned in the top left corner of XDX.
2 XFA presents upper right to XCA.
3 XCL is over there and XK is on the right of it.
4 The object XBZ is upper and slightly to the right of the object XFA.
5 XDX is to the upper left of XCL.
What is the relation of the agent XK to the agent XCA?

**Predicted Label**: right
**Ground Truth**: right | **Story**:
1 XCC and XBT are both there with the object XCC above the object XBT.
2 XDC is positioned left to XBT.
3 If XFK is the center of a clock face, XCR is located between 2 and 3.
4 XD presents upper right to XCC.
5 XFK presents over XDC.
What is the relation of the agent XCR to the agent XD?

**Predicted Label**: lower-left
**Ground Truth**: left |

Table 17: Correctly and Incorrectly Classified Instances by o4-mini

| $k$ | Correctly Classified Instances | Incorrectly Classified Instances |
|---|---|---|
| 1 | **Story**:
1 XGU is at XAN's 12 o'clock.
What is the relation of the agent XGU to the agent XAN?

**Predicted Label**: above
**Ground Truth**: above | **Story**:
1 XGD is positioned in the front right corner of XA.
What is the relation of the agent XGD to the agent XA?

**Predicted Label**: lower-right
**Ground Truth**: upper-right |
| 2 | **Story**:
1 The object XU is positioned directly below the object XR.
2 XAN and XR are next to each other with XR on the right and XAN on the left.
What is the relation of the agent XAN to the agent XU?

**Predicted Label**: upper-left
**Ground Truth**: upper-left | **Story**:
1 XAJ is over there with XFM below.
2 XJR is positioned in the top left corner of XFM.
What is the relation of the agent XJR to the agent XAJ?

**Predicted Label**: lower-left
**Ground Truth**: left |
| 3 | **Story**:
1 XJM is positioned above XU and to the right.
2 XJI is diagonally to the bottom right of XJM.
3 XIU is on the left and XJI is on the right.
What is the relation of the agent XIU to the agent XU?

**Predicted Label**: right
**Ground Truth**: right | **Story**:
1 XAW is sitting at the 12:00 position to XFQ.
2 XGZ is below and to the left of XHA.
3 XGZ is slightly off center to the top left and XFQ is slightly off center to the bottom right.
What is the relation of the agent XHA to the agent XAW?

**Predicted Label**: upper-left
**Ground Truth**: above |
| 4 | **Story**:
1 XJI is at the bottom of XBE and is on the same vertical plane.
2 XDP is to the top right of XBE.
3 XHY is there and XEN is at the 2 position of a clock face.
4 XDP is positioned above and to the left of XHY.
What is the relation of the agent XEN to the agent XJI?

**Predicted Label**: upper-right
**Ground Truth**: upper-right | **Story**:
1 XFO is on the top of XIZ and is on the same vertical plane.
2 XCL and XHP are both there with the object XCL above the object XHP.
3 XJR and XIZ are parallel, and XIZ is to the right of XJR.
4 XCL presents upper right to XFO.
What is the relation of the agent XHP to the agent XJR?

**Predicted Label**: right
**Ground Truth**: upper-right |

| $k$ | Correctly Classified Instances | Incorrectly Classified Instances |
|---|---|---|
| 5 | **Story**:
1 XBY is on the right side and below XEI.
2 XIU is positioned down and to the left of XF.
3 XF presents over XEN.
4 XEN is placed at the lower left of XJA.
5 XBY and XJA are in a vertical line with XJA below XBY.
What is the relation of the agent XIU to the agent XEI?

**Predicted Label**: lower-left
**Ground Truth**: lower-left | **Story**:
1 XAZ is north east of XIO.
2 XIO is on the right side to XBS.
3 XHL is positioned below XAZ and to the right.
4 XDR and XHL are parallel, and XDR is on top of XHL.
5 The objects XBS and XEG are over there. The object XBS is lower and slightly to the left of the object XEG.
What is the relation of the agent XDR to the agent XEG?

**Predicted Label**: upper-right
**Ground Truth**: right |

## K  Failure Modes

We include here the full figures for the analysis of the failure modes in the symbolic translation experiment detailed in Section 4.5. Figure 11 provides the full breakdown of the different types of semantic errors across all values of $k$ and all models used. Figure 12 presents the full breakdown of fact-translation errors, which can be seen as a further breakdown of the types of syntactic errors in Figure 11, since having too few facts (see the yellow and red bars) will guarantee an incorrect answer.

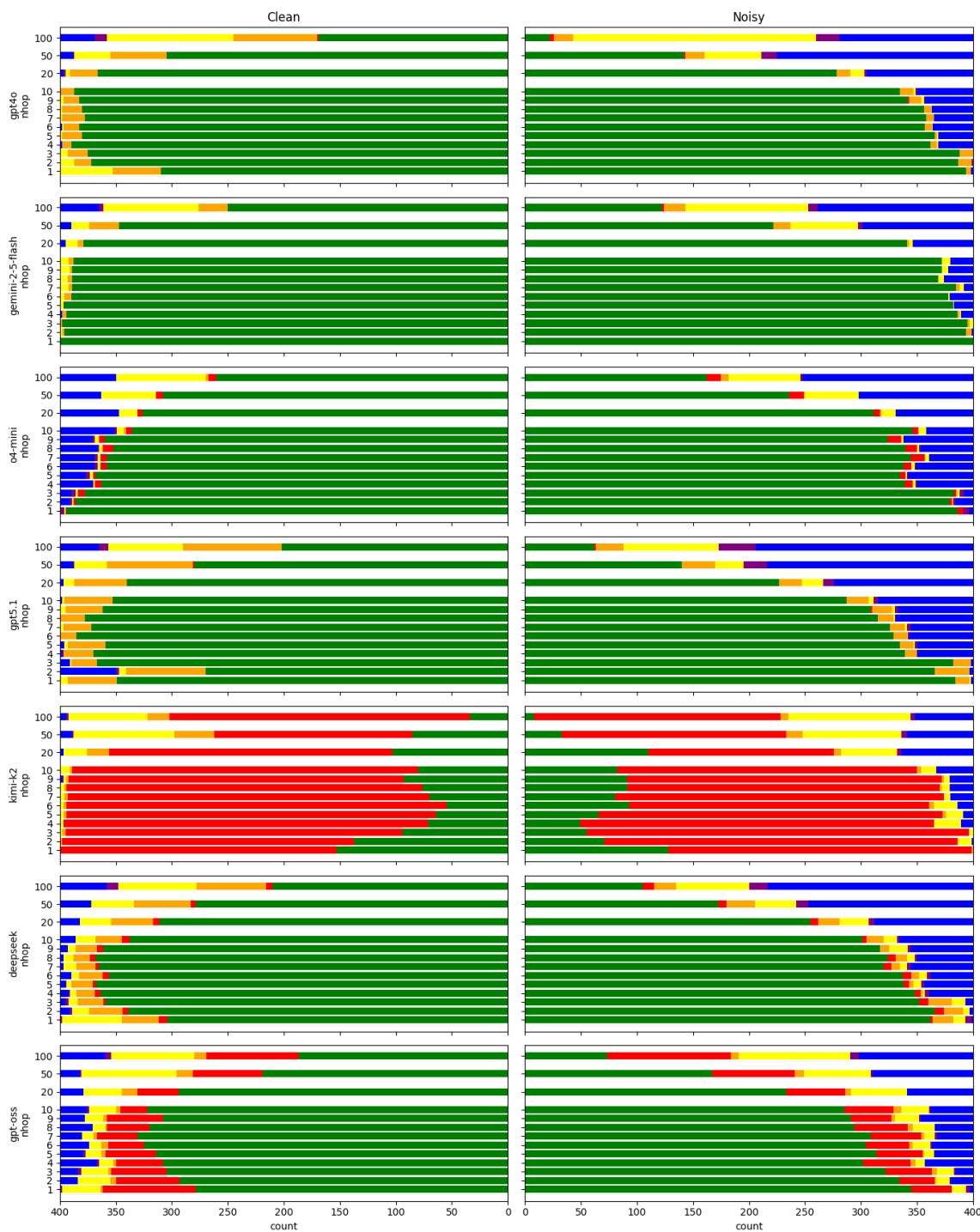

Figure 11: Full breakdown of syntactic vs semantic errors in the experiment in Section 4.5 by model, number of hops and noise. As in Figure 9, the colour-coding shows green: correct, red: syntactic error, orange: single incorrect answer, yellow: no answer, purple: multiple incorrect answers, blue: multiple answers of which at least one is correct.

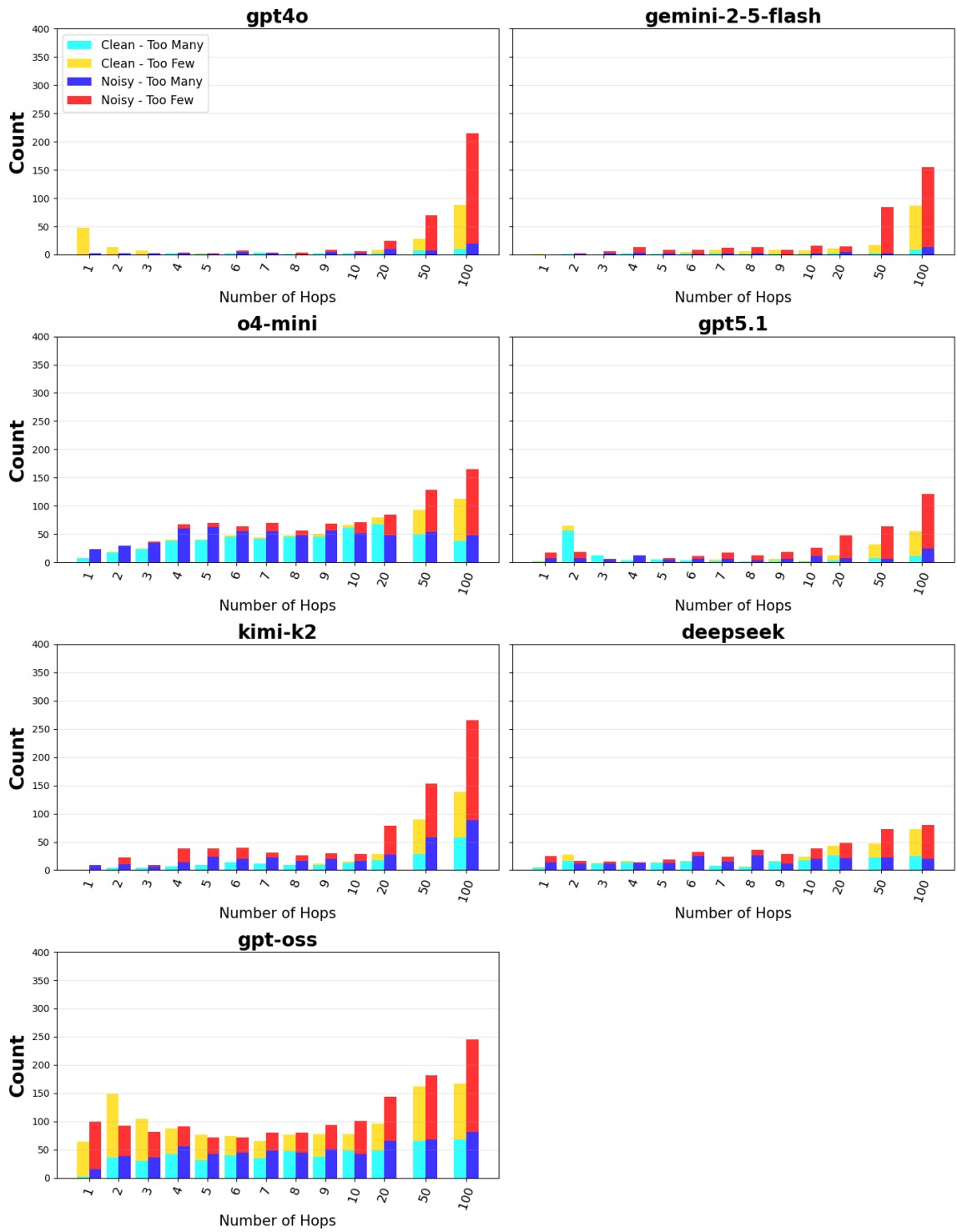

Figure 12: Counts for all models used in ASP-translation experiment in Section 4.5 of ASP-facts, colour coded by noise in blues vs reds for too few vs too many facts generated, and in dark vs light shades indicating noisy and clean data points.

