# OpenReview forum: "$\texttt{DecompSR}$: A Dataset for Decomposed Analyses of Compositional Multihop Spatial Reasoning"
_TMLR — Accepted by TMLR_

### Review · Reviewer_24Wv · 2026-02-07

**Summary Of Contributions:**

DecompSR is a verifiable generation framework for producing synthetic spatial reasoning tasks. Specifically aims to evaluate compositional spatial reasoning in the multiple facets of compositionality introduced by [1]. The framework produces triples of stories, questions, and answers, where the story describes nodes in a 2D grid and their spatial relations to each other, and the question asks for the spatial relation between two nodes. There are various results including benchmarking models on various number of hops, changing the order of the story, introducing distractors, changing node names, changing the language and evaluating their ability to translate a story into a symbolic program. Further analysis demonstrates that there is not a significant amount of correlation between the incorrect responses and that there can be a difference between syntactic and semantic errors in the symbolic translation experiment.

Strengths:
- Presents a coherent argument for the requirement of a compositional benchmark that is fully verifiable.
- Creatively address each of the 4 facets of compsitionality in the generation framework

Weaknesses:
- Only evaluating gpt-4o for the systematicity, overgeneralization, and natural language translation experiments is limiting. In the Substitutivity experiment, o4-mini has a clearly different trend than 4o, so we might expect o4-mini to have different trends in the other experiments as well. Not evaluating reasoning models for these tasks that are testing for different aspects of reasoning appears unproductive.
- No human baseline provided - it is difficult to say if this benchmark is evaluating spatial reasoning in a manner that is useful to humans. For example, can a human even solve a k=100 problem?
- Inconsistencies between the productivity experiment and the symbolic translation experiment - they use different suites of models and the trends between the two experiments don't line up. Thus, the hypothesis at the end of the section 4.6: "If models cannot reliably translate stories they are perhaps unlikely to accurately reason about them." is unsupported, and the purpose of this experiment at all can be questioned.
- No example outputs are provided - this would have greatly aided the analysis of failure modes. For example, the observation "we notice that as the reasoning depth increases, language models tend to guess randomly while reasoning models start abstaining" is unsupported. Furthermore, the few-shot prompt discourages any reasoning before outputting an answer - this may be problematic but without example outputs we can't tell.

[1] Dieuwke Hupkes, Verna Dankers, Mathijs Mul, and Elia Bruni. Compositionality Decomposed: How do Neural Networks Generalise? Journal of Artificial Intelligence Research, 67:757–795, 2020.

**Audience:**

No

**Audience Explanation:**

No, although spatial reasoning in general is widely interesting to the community, I believe that the benchmark presented is too synthetic and any trends stated are not substantive enough to be of interest. A lack of human baseline and analysis of example outputs makes it difficult to justify the relevance of the benchmark in addressing open problems in the area of spatial reasoning.

**Claims And Evidence:**

No

**Claims Explanation:**

No, the claim that this benchmark is evaluating multi-hop reasoning is not fully substantiated without example outputs clearly distinguishing success and failure cases. The inconsistencies between models used for evaluation across various experiments makes it more difficult to determine any common trends, or if the experiments are valid in the context of the suite of models available in recent times. Some conclusions drawn, such as the ability for a model to perform symbolic translation correlates with it's ability to do multi-hop reasoning are not numerically justified.

**Requested Changes:**

- Evaluate o4-mini for the systematicity, overgeneralization, and natural language translation experiments, or some other reasoning model.
- Evaluate some kind of human baseline.
- Align the suite of models between the productivity experiment and the symbolic translation experiment to draw a better conclusion of their relationship.
- Provide a few example outputs that are revealing - I'm genuinely curious about how exactly the models are failing for large k values and I'm sure other readers would be as well.
- Why is o1 so much worse than the rest of the models? This is counterintuitive, and so it would be helpful to highlight the cause for deeper understanding.
- In Figure 9, it is unclear to me how the orange, purple, and blue results arise.
- In Figure 10, the y-axes appear to be reversed, as some models get nearly perfect performance for k=100.
- Minor: few grammar issues across the paper, including in the abstract itself. Would be helpful to readers if section 2.2 explained spatial reasoning in depth and what that entails, instead of just an overview of benchmarks for testing it.

---

> ### Author Response · Authors · 2026-05-20
> **Response to reviewer 24Wv**
>
> ## (1/3)
> We thank the reviewer for the thoughtful and rigorous review. We appreciate the reviewer highlighting the utility of a verifiably correct dataset and are grateful for pointing out the creativity in our approach.
>
> We begin our rebuttal by addressing the reviewer’s list of “weaknesses” of the submission.
>
> > Only evaluating gpt-4o...
>
> We have now included results from`o4-mini` in each experiment in the submission (Tables 1, 2, appendices E, F, H) to meet the reviewers request. Of course it is desirable to test a large suite of models, but we have a limited budget so our choice of models is limited, something pointed out by reviewer xPmb. The main contributions of the submission are the DecompSR dataset and its generation method; the benchmarking indicates that the dataset is not stale or saturated as even very recent models like `gpt-5` perform at guess-rates for sufficiently complex DecompSR instances.
>
> We also note that productivity is the most salient aspect of compositionality (for instance [1,2,3] all claim that problems requiring longer reasoning chains are harder), which led to the more extensive benchmarking of productivity than other aspects (this is also noted in section 2.2 in  the submission). In developing DecompSR we corrected issues with similar spatial reasoning datasets [4,5] which report incorrect numbers of hops in their findings, and in the datasets themselves.
>
>
> >No human baseline provided...
>
> We appreciate that including a human baseline for comparison is interesting but it falls firmly outside the scope of both this paper and journal and should instead be asked in an experimental psychology, or cognitive science journal. The central idea behind DecompSR is to stress-test the different aspects of compositionality in language models. A human baseline, while interesting for studying limits of human compositional abilities, is not relevant to stress testing a machine's compositional abilities. It is like comparing a human's ability to perform long arithmetic calculations to a calculator's. Of course a human would not perform well on such tasks, in the same way a human may not perform well on DecompSR, but the point of the experiment was never to compare a machine’s ability to a human’s. Note also that there are many datasets accepted by the community that were published with no human baseline, presumably because it is not the purpose of these datasets to begin with.
>
> Beyond scope, we note that running a rigorous human baseline experiment is non-trivial. Sourcing representative participants is difficult and crowdsourcing platforms such as Amazon Mechanical Turk are well-documented to suffer from incentive misalignment, low attention, and non-representative samples (for e.g., please refer [7] for a detailed discussion on data quality related issues on MTurk). More fundamentally, the reproducibility crisis in experimental psychology [6] raises serious questions about the reliability of any such results, which is a concern that applies even to well-funded, carefully designed studies. We therefore do not believe that a human baseline, even if conducted, would add reliable scientific value to this submission.
>
> The core contribution of the submission is DecompSR and its verifiably correct generation method which makes this dataset a clear and simple task to evaluate the compositional abilities of LLMs, as highlighted by all other reviewers. As errors in LLM evaluation datasets are prominent and difficult to identify, DecompSR and its generation method stands out — a fact that all reviewers have notedas a novel contribution.
>
>
> > Inconsistencies between the productivity experiment and the symbolic translation experiment
>
> We ask the reviewer to reconsider the claim that there are inconsistencies between the symbolic translation and productivity experiments. The symbolic translation experiment uses a subset of the models of the productivity experiment (bar `gpt-OSS`), granted, we should clarify this is due to budget constraints. The statement at the end of 4.6 asks how a model is meant to be able to reason about a problem if it cannot even reliably abstract it. The symbolic translation experiment demonstrates empirically that models fail to translate even small numbers of sentences accurately, which serves as a probe of models’ abstraction abilities.
>
>
> > No example outputs are provided
>
> We have now included example output in Appendix J. See the next response for further motivation.

---

> ### Author Response · Authors · 2026-05-20
> **Response to reviewer 24Wv**
>
> ## (2/3)
> ##### Accuracy of claims:
> We would like to contest that the claims made in our paper are not accurate.
>
> > not fully substantiated
>
> We appreciate that presenting example output clarifies the claims we have made, so we have included examples of correct and incorrect output in the paper (Appendix J). However we would like to point out that all other reviewers disagree that this claim is not fully substantiated. Moreover the reviewer’s focus on human baseline suggests perhaps that the reviewer has read our claim as a cognitive scientific or experimental psychological one which is not the claim made in our paper as we focus on developing a dataset for stress-testing compositional abilities of language models.
>
> > determine any common trends
>
> We have not made claims about models’ compositional abilities overall, but rather in its different facets that DecompSR allows you to inspect. We have clearly determined common trends among the models used, in each of the subsections of Section 4 (pages 6-10). Note also that reviewer zsuh highlights this fact as a contribution of the submission.
>
>
> > makes it more difficult to determine if the experiments are valid in the context of the suite of models available in recent times.
>
> We are unsure what is meant by 'validity' in this context, as the dataset is verifiably correct, and the models used are contemporary ones. We would like to understand in closer detail what it is the reviewer thinks invalidates these experiments.
>
>
> ##### TMLR audience reception
>
> We challenge that *no one* (as stated by the reviewer) in the TMLR audience would be interested in the findings in submission. In fact all other reviewers have raised their substantial interest in this paper.
>
> The reviewer again highlights the lack of human baselines as a reason for why the TMLR audience would not be interested in this paper. We contest that such results are relevant to this submission and refer to our above comments on why this falls outside the scope of TMLR.
>
> ##### Requested changes
>
> > Evaluate `o4-mini`
>
> We have met this request, see in the updated submission results for o4-mini in all experiments.
>
> > Evaluate a human baseline
>
> See above.
>
> > Align the suite of models by evaluating o4-mini on systematicity, overgeneralisation and NL translation experiment
>
> We have included o4-mini results in these experiments in the revised submission.
>
>
> > Provide example outputs.
> We have done as requested in Appendix J.
>
> > Why is `o1` so much worse?
>
> We generated the confusion matrix of o1 model for nhops = 2 and nhops = 10 for comparison:
>
> ### nhop=2
> |p\t|lr|ur|ul|ll|l|b|r|a|o|
> |---|--|--|--|--|--|--|--|--|--|
> |lr|61|0|15|0|1|16|13|3|0|
> |ur|1|70|0|11|1|1|12|13|0|
> |ul|3|0|54|0|19|1|0|10|0|
> |ll|0|4|0|59|13|8|0|2|0|
> |l|0|0|4|3|32|3|3|2|0|
> |b|2|0|0|1|2|42|0|0|0|
> |r|5|0|0|0|2|0|37|0|0|
> |a|0|1|2|1|0|4|0|35|0|
> |o|3|0|0|0|5|0|10|10|0|
>
> ### nhop=10
> |p\t|lr|ur|ul|ll|l|b|r|a|o|
> |---|--|--|--|--|--|--|--|--|--|
> |lr|5|0|1|1|1|3|1|0|0|
> |ur|0|1|0|1|2|0|0|0|0|
> |ul|3|0|2|1|4|0|0|0|0|
> |ll|3|0|0|3|1|1|3|3|0|
> |l|0|0|0|2|0|2|0|2|0|
> |b|1|0|0|3|2|8|1|0|0|
> |r|4|1|0|0|1|1|0|5|0|
> |a|0|0|2|0|0|0|3|2|0|
> |o|59|73|70|64|64|60|67|63|0|
>
> Here each row represents the predicted (p) label where each column represents the true (t) label. lower-right(lr), upper-right(ur), upper-left(ul), lower-left(ll), left(l), below(b), right(r), above(a), other(o). Some examples of ‘other’ category include responses such as: ‘Cannot be determined from the given information.’, ‘No information is given to determine their exact relative position.’, ‘We cannot determine the exact relation from the story’. As the number of hops increases, a larger proportion of predictions fall into this “other” category, indicating growing uncertainty with reasoning depth.
>
>
> > Figure 9 colour coding:
>
> We have added a comment about what constitutes a correct answer in the beginning of section 4.6 and have illustrated examples of each colour coding in section 5.1 in reference to Figure 9.
>
>
> > Figure 10 y-axis
>
> We thank the reviewer for pointing this out. This was indeed an error in the plot which we have fixed in the revised submission.
>
> > Minor
>
> We have now provided a further introduction to spatial reasoning, see the addition in the existing introduction to spatial reasoning in section 2.2.
>
> —
>
> In conclusion, we again thank the reviewer for their comments. We hope our responses and amendments will encourage the reviewer to reconsider their comments to reflect our perspective.

---

> ### Author Response · Authors · 2026-05-20
>
> ## (3/3)
>
> [1] Lake, Baroni (2023) Human-like systematic generalization through a meta-learning neural network
>
> [2] Lake, Baroni (2018) Generalization Without Systematicity: On the Compositional Skills of
> Sequence-to-Sequence Recurrent Networks
>
> [3] Zhou, Denny, et al. (2022) Least-to-most prompting enables complex reasoning in large language models.
>
> [4] Shi, Zhang, Lipani (2022)  StepGame: A New Benchmark for Robust Multi-Hop Spatial Reasoning in Texts
>
> [5] Li, Cohn, Hogg (2024) Advancing Spatial Reasoning in Large Language Models: An In-Depth Evaluation and Enhancement Using the StepGame Benchmark
>
> [6] Ioannidis, J. P. (2005). Why most published research findings are false. PLoS medicine, 2(8), e124.
>
> [7] Webb, M. A., & Tangney, J. P. (2024). Too good to be true: Bots and bad data from Mechanical Turk. Perspectives on Psychological Science, 19(6), 887-890.

---

### Review · Reviewer_xPmb · 2026-02-17

**Summary Of Contributions:**

(Motivation) There is a rich history of evaluating compositionality in machine learning models. In bucket A, Benchmarks like ARC-AGI and GSM8k measure performance of LLMs on tasks that require reasoning, but avoid breaking these tasks down into sub-tasks that provide more insight about systematic abilities. In bucket B, work like SCAN and PCFG SET (Hupkes et al) test spatial reasoning systematically across well defined dimensions of compositionality, but their use of train/test splits and lack of natural language presentation is not directly applicable to LLMs. The DecompSR dataset proposed in this paper evaluates LLMs across 4 of the 5 dimensions of compositionality from PCFG SET (bucket B), but uses an in-context learning approach with natural language presentation (bucket A) that allows it to apply directly to evaluate modern LLMs.

(Approach) The DecompSR dataset is a natural language QA dataset (and dataset generator) about spatial relations between entities where the context for each QA pair is created by performing a random walk on a 2d grid. Varying the length of the walk, naming of the spatial objects and relations differently, translating the QA pairs to Swedish and Hindi, changing the order of presentation, and adding distractor information allows the dataset to systematically vary the 4 dimensions of compositionality from PCFG SET. To quote the paper, these are "productivity (reasoning depth), substitutivity (entity and linguistic variability), overgeneralisation (input order, distractors) and systematicity (novel linguistic elements)." All QA pairs are verified using an oracle solver implemented in ASP.

(Experiments) Experiments evaluate many LLMs including GPT-5, Claude Sonnet 3.7, and Gemini 2.5-pro, but mostly focusing on GPT 4o. They show
* (Productivity) Performance decreases eventually to near random as the number of reasoning hops required exceeds 50 or 100.
* (Systematicity) Using nonsense words, Hindi, and Swedish results in modest performance drops over English.
* (Overgeneralization) Adding distractor noise and shuffling context presentation has a minor impact on performance.
* (Substitutivity) Replacing entity and relation names has a minor impact on performance.
* (Tool) Using the same LLMs to parse the problems into logical statements and evaluating those with an ASP solver results in improved performance.

**Audience:**

Yes

**Audience Explanation:**

I think this paper will be used for the benchmark itself, which is a more careful, systematic and fine-grained evaluation than many common benchmarks, though it is also narrow. Thus it has some relevance for those looking to evaluate the latest LLMs.

The results reported in the paper are less relevant because of the concern about tool calling and reasoning models expressed below.

**Claims And Evidence:**

Yes

**Claims Explanation:**

* "a novel methodology and open-source framework for generating correct-byconstruction natural language spatial reasoning tasks that enable decomposed evaluation of compositional abilities beyond mere accuracy; the release of DecompSR, a large-scale, customisable benchmark dataset (over 5 Million samples) for multi-hop compositional spatial reasoning;"
    * This is supported. The work is novel, open source (including anonymous links to data and code that were available from the pdf), and its correctness well supported by the dataset's construction and use of ASP oracle.
* "comprehensive benchmarking of contemporary LLMs, revealing specific strengths and weaknesses in their systematic reasoning capabilities;"
    * The evidence toward this claim could be improved by including more experiments with slightly more recent LLMs, but the claim is mostly true, with many recent LLMs evaluated including reasoning models and (essentially) tool calling.
* "empirical findings demonstrating that while LLMs show some linguistic resilience, they largely struggle with systematic and productive generalisation in spatial tasks and exhibit overgeneralisation tendencies."
    * This is well substantiated by the experiments across all 4 dimensions, though perhaps the impact of overgeneralization is a bit less than implied by this claim.

**Requested Changes:**

(Relevance giving tool calling and reasoning) Tool calling and reasoning models are important modern approaches for addressing generalization and compositionality of the sort evaluated here. This paper evaluates some reasoning models and has a tool calling style experiment (section 4.6), but I don't think it is representative of the state of the art in either approach. For example, what would happen if this problem were given directly to a coding agent (e.g. Opus 4.6). How does performance of a reasoning model (e.g., GPT-5) perform across different budgets (e.g., "medium", "high", "xhigh")? Providing experiments toward these would help make the results more relevant. It would also be good to acknowledge the ways the paper does address these concerns more directly in the introduction.
* A related concern is that the reasoning levels used for the GPT-5 and Gemini 2.5-Pro experiments is not disclosed in the paper. That detail should be added.
* I'm sure part of the reason this is less well explored is because of the cost of doing those experiments. In that vein it would be useful to describe the cost of each set of experiments and possibly report some results on a cost-accuracy pareto curve.

Figure 1 should include visualizations of all the manipulations described instead of just some of them. (e.g., add re-ordering and distractors)

---

> ### Author Response · Authors · 2026-05-20
> **Response to reviewer xPmb**
>
> We thank the reviewer for their thorough response, in particular for highlighting the novelty of our approach using symbolic verification to ensure correctness of the dataset. That the reviewer also emphasises the dataset is the main contribution of this work is particularly welcome, as this was the core contribution in mind when developing DecompSR.
>
> Regarding the requested changes:
>
> > Tool-calling/agentic methods
>
> As you have noted, DecompSR serves as a dataset for those interested in evaluating compositional reasoning abilities of systems. This constitutes the core contribution of the paper. The benchmarking we performed already covers a broad range of LLMs and, as you point out, some tool calling approaches use Answer Set Programming. As you already acknowledge,we have a limited budget. We think the benchmarking we have performed is sufficient to show that DecompSR is a challenging dataset (note for example GPT 5’s poor performance for k=100) and that DecompSR should instead be benchmarked by more advanced systems in future work.
>
>
> > Disclose reasoning levels for GPT-5 and Gemini 2.5 pro
>
> All reasoning models were implemented using default settings. For GPT-5, the default reasoning level is medium and for Gemini 2.5 pro uses a dynamic thinking budget as default.  Note that we used GPT-5.1 which does not have adaptive reasoning, as opposed to its recent successors.
>
>
> > Disclose cost of experiments
>
> The total spend was $3900. See Appendix C.1 for a detailed breakdown.
>
> > Figure 1 should include visualizations of all the manipulations described instead of just some of them. (e.g., add re-ordering and distractors)
>
> We thank the reviewer for this suggestion. We have now updated Figure 1 to include all possible manipulations of DecompSR.

---

### Review · Reviewer_zsuh · 2026-05-04

**Summary Of Contributions:**

The paper revolves around large-scale benchmark dataset, DecompSR. The compositional spatial reasoning abilities of Large Language Models has been rigirously evaluated. This framework provides users with the chance to independently manipulate specific variables to evaluate four distinct facets of compositionality:

1. productivity (the number of reasoning steps),
2. systematicity (the use of linguistic elements),
3. substitutivity (varying entity names and natural language),
4. overgeneralization (adding distractors).

The authors show that while LLMs exhibit some resilience to linguistic variations, they significantly struggle with productive and systematic generalization as tasks become more complex

**Audience:**

Yes

**Audience Explanation:**

It is a novel perspective to generate the datset via mathematical verifier (symobolicaly)

**Claims And Evidence:**

Yes

**Claims Explanation:**

DecompSR is algorithmically generated and mathematically verified using an Answer Set Programming symbolic solver. this pivots more around model's reasoning deficits rather than errors, ambiguities, or label noise in the dataset. By controlling parameters like reasoning depth, language variation, entity types, and the presence of noise, they force models to engage with the deep compositional structure of the problem

**Requested Changes:**

It would have been better if the focus would be less on non-english languages and if it is meant to include non-english languages it is better to be expanded. It also would have been better if it wasn't restricted to 2D spatial reasoning and one could further see the grounding impact, although this part is not very critical. Many of results are in fact after references. It would be better if these results would move up to main body. The results pivot largely on the closed-source models. It would be better if more open-source models have been added. In table 4, some of the models drop sharper than the other one when k changes from 1 to 2. It would be better if explanation for this has been provided.

---

> ### Author Response · Authors · 2026-05-20
> **Response to reviewer zsuh**
>
> We thank the reviewer for the positive review, especially how it underscores the algorithmic generation of DecompSR and how this lets users generate further data to their own wishes.
>
> Regarding the requested changes:
>
> > It would have been better if the focus would be less on non-english languages and if it is meant to include non-english languages it is better to be expanded.
>
> We included the linguistic variation as a supplementary experiment to test multilingual abilities of these models and to showcase the modular nature of the DecompSR codebase. This experiment is not mentioned as a core contribution of the paper, but we do agree that we should include further detail into this aspect of DecompSR, and have done so in Appendix I.
>
> > It also would have been better if it wasn't restricted to 2D spatial reasoning and one could further see the grounding impact, although this part is not very critical.
>
> We would like to highlight that the decision of working in 2D was made in order to give tighter control of the structure of the data. Moreover, many domains only require 2D and thus it constitutes a useful and important kind of spatial reasoning in itself. This being said, we are considering extensions of this work in settings grounded in visual data as future work, as well as in 3D.
>
> > Many of results are in fact after references. It would be better if these results would move up to main body.
> We appreciate this point and have made efforts to incorporate more of the appendix materials into the body of the paper. In particular we have incorporated more results of all the experiments into the main body of the paper (Table 1, Table 2, Figure 10).
>
> >The results pivot largely on the closed-source models. It would be better if more open-source models have been added.
>
> This would absolutely make the benchmarking richer, and while we have gone to some lengths to include more models in the benchmarking in our revised submission, our finances have put hard constraints on this ability. However, since the main contribution of this paper is the dataset itself, and its generation method, we would expect the community to end up providing further benchmarking as people start evaluating their models on DecompSR.
>
> > In table 4, some of the models drop sharper than the other one when k changes from 1 to 2. It would be better if explanation for this has been provided.
>
> We have provided a detailed explanation of this in Table 8 in Appendix D. Please see the paragraph highlighted in Appendix D for full detail. In short, the decline in accuracy at k=2 depends upon multiple factors, including the recency of the model, whether the model is designed as a general-purpose or explicit reasoning-oriented system, and the organization responsible for developing the model.

---

### Review · Reviewer_FWw7 · 2026-05-18

**Summary Of Contributions:**

The authors introduce a large, procedurally generated benchmark dataset for analyzing compositional spatial reasoning in LLMs. The dataset is correct by construction, verified via a symbolic solver, and allows independent control over key aspects of compositionality such as productivity and substitutivity. Furthermore, the paper benchmarks LLMs, revealing struggles with productive and systematic generalization but robustness to linguistic variation.
The work’s primary strengths lie in its rigor and novelty. The dataset’s procedural generation and symbolic verification ensure correctness and fine-grained control over compositionality dimensions, while the comprehensive evaluation across LLMs provides actionable insights into their spatial reasoning limitations. However, the paper has notable weaknesses. The motivation for why spatial reasoning over natural language is a necessary or interesting direction for LLMs is weak, and it does not address why LLMs should tackle this directly rather than mapping text to geometric/graph representations plus logical solvers. The focus on 2D spatial reasoning feels arbitrary and unchallenging, as 3D or map-based tasks would better test generalization and real-world applicability. Additionally, some figures, particularly Figures 8 and 9, are poorly curated with text so tiny it is difficult to read, which detracts from the paper's clarity.

**Audience:**

Yes

**Audience Explanation:**

The dataset’s rigor and modular design are valuable for researchers studying compositional reasoning in LLMs, and the findings about struggles with productivity and systematicity align with current debates in the field. However, the paper misses opportunities to connect spatial reasoning to broader LLM capabilities, such as planning or multi-step reasoning. A stronger motivation section would clarify why this matters beyond the benchmark itself. The focus on spatial reasoning may limit interest to a subset of TMLR’s audience, such as those working on logical reasoning, neuro-symbolic systems, or benchmark design.

**Broader Impact Concerns:**

No major ethical concerns are apparent, but the paper should address potential misuse. If DecompSR is used to train LLMs for spatial reasoning in high-stakes domains like robotics or navigation, errors could have real-world consequences. A Broader Impact Statement should acknowledge this and discuss mitigation strategies, such as hybrid LLM-symbolic systems. Additionally, the procedural generation of the dataset might inadvertently encode biases in spatial relationships or linguistic patterns, so the authors should verify fairness across demographic or cultural variations in spatial reasoning.

**Claims And Evidence:**

Yes

**Claims Explanation:**

The claims about LLMs struggling with productive and systematic generalization are mostly well-supported by the extensive benchmarking. However, such benchmark has its own limitations. For example, the k=50 hop experiments are unrealistic (as humans would fail here too) and may skew the narrative; reducing k to 10 or 20 would make the findings more interpretable and actionable. The substitutivity tests using nonsensical node names are methodologically flawed, as LLMs inherently rely on textual representations, so testing their robustness to meaningless linguistic variations does not yield meaningful insights about reasoning ability. Therefore, these experiments should be removed or rethought. Finally, the linguistic variation tests also lack justification, as it is unclear what additional insights they provide beyond confirming LLMs’ known sensitivity to input phrasing.

**Requested Changes:**

The following issues should be addressed to improve the paper:
- Clearly explain why spatial reasoning over natural language is a necessary or interesting direction for LLMs, and address the alternative of using LLMs to map text to geometric/graph representations plus logical solvers (e.g., ASP); why is end-to-end LLM reasoning preferable here?
- Remove or rework the substitutivity tests with nonsensical names or directions, as they do not effectively test reasoning ability.
- Cap the value of k at 10 or 20 hops and re-run experiments, as the current k=50 is unrealistic and obscures practical insights.
- Expand the related work section to discuss neuro-symbolic approaches like DeepProbLog [1], NeurASP [2] or others [3,4] as baselines or alternatives.
- Better curate the figures, particularly Figures 8 and 9, to ensure text is legible.
- Clarify what new insights the language variation experiments provide, as the current justification is unclear.
- Consider adding 3D or map-based reasoning tasks as an extension to demonstrate the dataset’s scalability and real-world relevance.
- Include ablation studies analyzing how individual compositionality dimensions interact with model performance.

[1]. Manhaeve, Robin, et al. "Neural probabilistic logic programming in DeepProbLog." Artificial Intelligence 298 (2021): 103504.

[2]. Yang, Zhun, Adam Ishay, and Joohyung Lee. "NeurASP: embracing neural networks into answer set programming." Proceedings of the Twenty-Ninth International Conference on International Joint Conferences on Artificial Intelligence. 2021.

[3]. Marra, Giuseppe, et al. "From statistical relational to neurosymbolic artificial intelligence: A survey." Artificial Intelligence 328 (2024): 104062.

[4]. Ciatto, Giovanni, et al. "Symbolic knowledge extraction and injection with sub-symbolic predictors: A systematic literature review." ACM Computing Surveys 56.6 (2024): 1-35.

---

> ### Author Response · Authors · 2026-05-20
> **Response to reviewer FWw7**
>
> ## (1/2)
> We thank the reviewer for the thoughtful and clear review. We in particular appreciate the reviewer highlighting the core contribution of this work, namely the dataset itself and the novelty and utility of its generation procedure. Furthermore, we appreciate the reviewer pointing out some of the preliminary insights we have found out about spatial reasoning limitations of LLMs through the benchmarking experiments.
> > Motivation for WHY spatial reasoning in NL is even necessary
>
> We will motivate this more clearly in the paper by including the following: “Spatial information, and in particular qualitative spatial information is ubiquitous in text and being able to reason about it is essential if computers are to be able to process it effectively. For example, consider the Corpus of Lake District Writings [1] a three-century long corpus of travel writings including works by Coleridge and Wordsworth which has rich spatial information in it. Equally, being able to reason about spatial information is key to understanding the everyday physical world, which humans (and robots) inhabit and perform actions located in time and space.  Being able to communicate about spatial information underlies this, and indeed much of common sense, e.g. see Hayes' Naive Physics Manifesto [2]. Further there is research demonstrating how linguistic spatial relations influences learning [4,5]. There is also a growing interest in studying the internal representations of spatial information in LLMs as studied in [6]” in section 2.2.
> > Does not motivate why LLMs should tackle decompSR at all
>
> We are happy to clarify this and have included the following motivation in the introduction of the paper: "Since LLMs see copious amounts and types of text including natural language spatial descriptions during pre-training, fine tuning and post-training, it is perhaps only logical to first focus on the predominant modality of LLM's training as a method to evaluate the compositional abilities of LLMs."
>
>
> >2D feels arbitrary (why not 3D? or map-based tasks)
>
> The 2D design of DecompSR allows us to control the different aspects of compositionality neatly and sufficiently quickly to get signal for future work. Moreover, many domains do need more than 2D information, and humans tend to abstract to 2D frequently (for example ignoring terrain when planning a route, at least initially. Pursuing even richer instances like working in 3D or in vision/map-based settings constitute the next steps for this work.
>
>
> >4. Poor figures (8 & 9 in particular)
>
> We have amended the small fonts in Figures 8 and 9 (now Figures 9 and 10) and taken steps to ensure remaining figures are legible such as removing the numerators in figure 8 and increasing font sizes where possible.
>
>
> > k≥50 experiments are unrealistic
>
> We clarify here and include in the motivation of the paper: "The core focus of DecompSR is to provide a reasoning task which stress-tests compositional abilities. It is desirable for an automated reasoning system to be robust in its reasoning even under stress of long reasoning-chains, novel components, systematically altered entity names and so on. decompSR provides a flexible and robust dataset to evaluate these abilities to extreme lengths."
>
>
> > Substitutivity experiments are flawed
>
> We notice the reviewer has not noted the use of the familiarisation prompt which effectively provides a translation between the new nonsense words and English. We hope this helps convince the reviewer that the experiment is not flawed.
>
>
> > Linguistic variation tests lack justification
>
> In the submission we make the point that the linguistic variation allows one to test substitutivity, although we understand this is confusing as we separate the language variation experiment from the "substitutivity experiment". To remedy this, we have included what was previously section 4.5 (Natural Language Translation Experiment) into section 4.4 (Substitutivity Experiment).
> > Expand the related work section to discuss neuro-symbolic approaches…
>
> We are happy to expand on this aspect of the work and have included a paragraph in the background section on compositionality. (We assume the reviewer means the background section when they talk of the related work section.)
>
> > Address potential misuse
>
> We have included a broader impact statement which acknowledges and addresses potential misuse of DecompSR at the end of the main body of the paper. We appreciate the reviewer raising this concern.

---

> ### Author Response · Authors · 2026-05-20
>
> ## (2/2)
> [1] Gregory, I., Donaldson, C., & Taylor, J. E. (2019). Landscape Appreciation in the English Lake District: A GIS Approach. Mapping Landscapes in Transformation: Multidisciplinary Methods for Historical Analysis (pp. 277–300)
>
> [2] Hayes, Patrick J. (2013)  The Second Naive Physics. Readings in Qualitative Reasoning About Physical Systems: 46
>
> [4] Carlson, Laura A. Logan, Gordon D. (2005) Attention and Spatial Language
>
> [5] Gilligan, Katie A. and Flouri, Eirini and Farran, Emily K. (2017) The Contribution of Spatial Ability to Mathematics Achievement in Middle Childhood
>
> [6] Wu, Wenya and Deng, Weihong (2025) Spatial Representation of Large Language Models in 2D Scene

---

### Decision · Action_Editor_Unyz · 2026-06-06

**Recommendation:** Accept as is

**Audience:**

Yes

**Audience Explanation:**

While the construction may feel synthetic and there can be other ways to solve the spatial reasoning tasks (e.g., through tool use and code generation), the main contribution of this paper is to propose a principled benchmark for evaluation of 2D spatial reasoning. In that respect, it will be interesting to researchers studying reasoning, who can utilize the dataset and extend the current results. The results evaluating different models are also useful, though I suspect they'll need to be updated as newer models keep being released.

**Claims And Evidence:**

Yes

**Claims Explanation:**

The paper proposes a benchmark for evaluating 2D spatial reasoning in LLMs. Compared to existing benchmarks, it provides a more fine-grained evaluation since it is procedurally generated and symbolically verified. The results evaluate different LLMs and show gaps in spatial reasoning across models such as GPT-4o and o4-mini. The analysis on multiple language (Hindi, Swedish) is a nice addition.

The revised paper helped address the concerns of the reviewers through experiments with more models and examples of failure cases.